# Specialization after Generalization: Towards Understanding Test-Time Training in Foundation Models

**Jonas Hübotter**[*,1]    **Patrik Wolf**[*,1,2]    **Alexander Shevchenko**[*,1]
**Dennis Jüni**[1]    **Andreas Krause**[1]    **Gil Kur**[1]

[1]ETH Zürich, Switzerland    [2]Max Planck Institute for Intelligent Systems, Tübingen, Germany

## Abstract

Recent empirical studies have explored the idea of continuing to train a model at test-time for a given task, known as test-time training (TTT), and have found it to yield significant performance improvements. However, there is limited understanding of why and when TTT is effective. Earlier explanations mostly focused on the observation that TTT may help when applied to out-of-distribution adaptation or used with privileged data. However, the growing scale of foundation models with most test data being in-distribution questions these explanations. We instead posit that foundation models remain globally underparameterized, with TTT providing a mechanism for *specialization after generalization*—focusing capacity on concepts relevant to the test task. Specifically, under the linear representation hypothesis, we propose a model in which TTT achieves a substantially smaller *in-distribution* test error than global training. We empirically validate our model's key assumptions by training a sparse autoencoder on ImageNet, showing that semantically related data points are explained by only a few shared concepts. Finally, we perform scaling studies across image and language tasks that confirm the practical implications of our model, identifying the regimes where specialization is most effective. Beyond TTT, our results provide additional strong evidence in support of the linear representation hypothesis.

## 1 Introduction

Since the "ImageNet moment" in 2012 when AlexNet won the ImageNet challenge (Krizhevsky et al., 2012), scaling data, parameters, and compute have led to foundation models, which are large-scale neural networks trained on vast datasets that can be applied across a wide range of use cases (Bommasani et al., 2021). This has spurred research on scaling laws, suggesting that scaling pre-training of a single model on a broad data distribution is sufficient for good performance on downstream tasks (Kaplan et al., 2020; Henighan et al., 2020; Hoffmann et al., 2022). With first-generation foundation models, fine-tuning was used primarily to adapt models to out-of-distribution test data (i.e., with a distribution shift) or to leverage fresh training data that was not seen during pre-training (so-called "privileged" data). Test-time training (TTT; Sun et al., 2020; Hardt & Sun, 2024; Akyürek et al., 2025) emerged as pushing this mechanism to the extreme: fine-tuning a separate model for each prediction. In recent years, foundation models have grown so large that most test data is effectively "in-distribution", meaning the model has encountered similar data during pre-training. This raises a key question:

> Can TTT improve predictions *even* in-distribution while using only already-seen data?

While foundation models typically have a large parameter count, extensive results on scaling laws show continuing improvements in performance when scaling model size (Kaplan et al., 2020; Bubeck & Sellke, 2021). Consequently, our work posits that today's foundation models are effectively "underparameterized", meaning that performance can be improved by further scaling. We hypothesize that due to this effective underparameterization, even if test data is in-distribution,

---

[*]Equal contribution. Correspondence to Jonas Hübotter `jonas.huebotter@inf.ethz.ch`.

the model cannot simultaneously approximate the ground truth across the full data distribution. TTT offers a mechanism to *specialize* the model to a local area around the test example. By temporarily "forgetting" irrelevant pre-trained knowledge, the model "frees up" capacity to learn the relevant concepts to the immediate task at a higher resolution. We refer to this mechanism as *specialization after generalization*. The mechanism of TTT—temporarily reallocating capacity by "forgetting" irrelevant knowledge—connects to concepts of capacity saturation and interference studied in continual learning (McCloskey & Cohen, 1989; Kirkpatrick et al., 2017).

We propose to model this phenomenon under the *linear representation hypothesis* (LRH; Mikolov et al., 2013; Park et al., 2024; 2025), which postulates that models represent high-level concepts—meaningful semantic features—as directions in a latent space. This latent concept space is typically assumed to be sparse conditioned on the input, meaning that any given input activates only a few possible concepts. For a particular prediction task, the linear coefficients of the active concepts effectively characterize their "meaning" for that task. Because the number of real-world concepts far exceeds a model's dimensionality, these concepts are superimposed within the model's dense activations (cf. Figure 1; Elhage et al., 2022). The LRH has been used extensively in prior work on interpretability (Kim et al., 2018) and activation

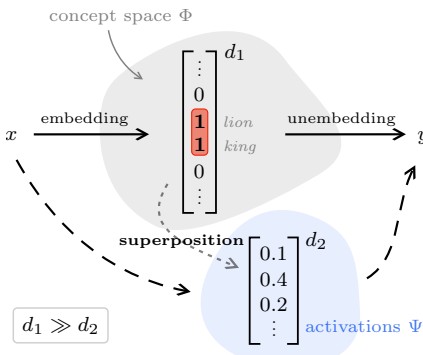

Figure 1: Setting: large concept space $\Phi$, superimposed on the feature space $\Psi$.

steering (Bolukbasi et al., 2016; Templeton et al., 2024) of foundation models. In this work, we analyze a model where TTT can learn the meaning of these superimposed concepts from data more efficiently than training a "global" model or non-parametric methods.

**Setting: The linear representation hypothesis.** The LRH can be formalized as follows. We assume the *existence* of an $s$-sparse *concept space* $\Phi : \mathcal{X} \to \mathbb{R}^{d_1}$ which is approximated by a learned *feature map* $\Psi : \mathcal{X} \to \mathbb{R}^{d_2}$ with $d_2 \ll d_1$. The ground truth to be learned is linear in the concept space, i.e., $f^\star(x) = \langle \Phi(x), w_\star \rangle$ for some unknown $w_\star \in \mathbb{R}^{d_1}$.[1] In Section 3, we leverage the LRH to develop a mechanistic understanding of *how* TTT behaves, and make the following key observations:

**O1:** The learned features $\Psi$ yield similar neighborhoods to neighborhoods in concept space $\Phi$.

**O2:** Among a test point $x^\star \in \mathcal{X}$ and its neighborhood (in $\Psi$-space), $f^\star$ can be approximated by an $s$-sparse linear function in the concept space $\Phi$.

**O3:** TTT in $\Psi$-space finds approximately the same task-specific model as sparse TTT in the concept space, indicating that TTT implicitly adjusts coefficients based on only a few concepts relevant to the test task.

Based on the LRH and our key observations, in Sections 4 and 5, we analyze *when* and *why* TTT can be effective. We find that **TTT improves predictions in the underparameterized regime, but its improvement diminishes as models become overparameterized.** We support this finding empirically and theoretically:

- **When? Implications for real data (§4):** We analyze TTT in a setting where we keep the learned superimposed concepts (i.e., $\Psi$) fixed and only update last-layer weights. Through scaling studies in image classification and language modeling, we find that TTT improves accuracy when the model is underparameterized, i.e., before the test loss saturates with increasing model size. Recent empirical findings also support this view (Lim et al., 2025; Doimo et al., 2024), showing that TTT learns the local meaning of concepts rather than discovering new concepts.

- **Why? Theoretical insights (§5):** We analyze the in-distribution test error of TTT in comparison to the test error of a globally trained model. Under the LRH and our key observations, we show that TTT generalizes at test-time even if the model is globally underparameterized (i.e., the feature space is exponentially smaller than the concept space: $d_2 \sim \log d_1$). In contrast, we show that if concepts are superimposed in an underparameterized feature space, the model cannot globally disentangle the meaning of all concepts.

---

[1]In classification, the logits are canonically parameterized as $\langle \Phi(x), w_c \rangle$ where $w_c$ is the class-specific weight vector. The class probability is then given by $\mathbb{P}(c \mid x) \propto \exp(\langle \Phi(x), w_c \rangle)$.

## 2 RELATED WORK

**TTT.** In the classical machine learning paradigm, models are trained on a fixed training set and then kept frozen during evaluation. Despite this standard practice that was used for decades, early work suggested specializing the model at test-time to each prediction task—such examples are local learning (Cleveland, 1979; Cleveland & Devlin, 1988; Atkeson et al., 1997) and local fine-tuning (Bottou & Vapnik, 1992). More recently, the idea of TTT (Sun et al., 2020; Wang et al., 2021) has regained attention in the context of fine-tuning large foundation models during evaluation (e.g., Krause et al., 2018; Hardt & Sun, 2024; Sun et al., 2025). TTT for a few gradient steps on (self-)supervised losses has since shown success in domains such as control (Hansen et al., 2021), abstract reasoning (Akyürek et al., 2025; Zweiger et al., 2025), language modeling (Hardt & Sun, 2024; Hübotter et al., 2025; Sun et al., 2025; Zhang et al., 2025; von Oswald et al., 2025; Yu et al., 2025), and video generation (Dalal et al., 2025). Many standard TTT methods train on carefully selected data from the pre-training dataset (i.e., do not add any new privileged information; Hardt & Sun, 2024; Hübotter et al., 2025), and several works studied how to optimally select data for imitation, e.g., the early seminal work of MacKay (1992) and recent extensions (Hübotter et al., 2024; Bagatella et al., 2025b). TTT has also been extended from supervised learning to reinforcement learning (Zuo et al., 2025; Bagatella et al., 2025a; Diaz-Bone et al., 2025).

So far it has not been well understood why and when TTT is effective. While many different methods have been proposed for TTT, we focus here on analyzing "semi-parametric" TTT (e.g., Hardt & Sun, 2024; Hübotter et al., 2025), where a pre-trained model is fine-tuned with a supervised loss on a small neighborhood of the test point in the training data. This is different from some other methods for test-time "adaptation", which are commonly applied with distribution shifts (e.g., Wang et al., 2021; Zhang et al., 2022; Durasov et al., 2025). Basu et al. (2023) consider a similar setting to ours, but analyze it through the lens of non-parametric estimation, relying on the smoothness of the target function in the feature space $\Psi$. In contrast, our framework explicitly models the underlying sparse concept space $\Phi$. This explains why TTT substantially outperforms "non-parametric" methods even when the function is locally high-dimensional ($s$-sparse) in the concept space. Furthermore, while most prior theoretical work simply assumes the TTT gradient aligns with the gradient on the oracle label (e.g., Sun et al., 2020), our work provides an idealized model where this alignment is justified.

**Sparse autoencoders (SAEs) and the LRH.** Our theoretical framework is built upon the LRH, which posits that foundation models represent high-level concepts as linear directions in their activation spaces. This idea has its roots in early word embedding models, which famously showed that semantic analogies could be solved with simple vector arithmetic (Mikolov et al., 2013; Pennington et al., 2014; Arora et al., 2016). More recently, the LRH has been validated across a wide range of models and domains, with studies identifying linear representations for abstract concepts like sentiment (Tigges et al., 2023), the state of a game board (Nanda et al., 2023), and even fundamental axes of space and time (Gurnee & Tegmark, 2024). A key tool for discovering and studying these conceptual directions are SAEs (Makhzani & Frey, 2014; Lieberum et al., 2024; Gao et al., 2025). SAEs are auxiliary models trained to reconstruct a foundation model's internal activations from a sparse, overcomplete dictionary of features. This process often yields features that are monosemantic, or aligned with single, human-interpretable concepts, thereby providing an empirical method to uncover the sparse concept space we consider in our work (Cunningham et al., 2024; Templeton et al., 2024).

## 3 *How* DOES SPECIALIZATION BEHAVE?

In this section, we begin by developing a mechanistic understanding of *how* TTT behaves. Since the "true" hypothesized concept space $\Phi$ is not accessible, we train SAEs to learn an approximate concept space $\hat{\Phi}$ whose properties can be analyzed. We use a top-$k$ SAE (Gao et al., 2025) to obtain sparse feature representations. After a brief description of the experimental setup, we detail our key observations **O1-O3** from Section 1.

### 3.1 EXPERIMENTAL SETUP

**SAE framework.** The SAE encoder projects a dense input vector $\Psi(x) \in \mathbb{R}^{d_2}$ to a learned high-dimensional, sparse representation $\hat{\Phi}(x) \in \mathbb{R}^{d_1}$:

$$\hat{\Phi}(x) := \text{top}_s(E \cdot \Psi(x)), \quad E \in \mathbb{R}^{d_1 \times d_2},$$

where the $\mathrm{top_s}$ operator retains the $s$ highest values and sets all others to zero. A linear decoder then reconstructs the original vector from this sparse representation:

$$\hat{\Psi}(x) := D \cdot \hat{\Phi}(x), \quad D \in \mathbb{R}^{d_2 \times d_1}.$$

The encoder $E$ and decoder $D$ (here for simplicity without bias terms) are optimized to minimize the reconstruction error:

$$\mathbb{E}_x \|\Psi(x) - \hat{\Psi}(x)\|_2^2 \to \min_{E,D}.$$

To mitigate the issue of "dead features" (elements of $\hat{\Phi}(x)$ that are never activated), we incorporate a ghost gradient auxiliary loss (Gao et al., 2025), which resulted in only $4\%$ inactive concepts in our experiments.

**ImageNet experiments.** We use the ImageNet-1K dataset (Deng et al., 2009). The dense vectors $\Psi(x)$ are normalized CLIP embeddings (Radford et al., 2021) of dimension $d_2 = 512$ (we use only the <CLS> component). We set the sparse dimension to $d_1 = 8 \times d_2 = 4096$ and the sparsity level to $s = 16$. For our analysis, we use the SAE's reconstructions $\hat{\Psi}(x)$ rather than the original CLIP embeddings $\Psi(x)$. This choice aligns our experiments more closely with our theoretical model and circumvents known challenges in training SAEs on raw, complex embeddings. This comes at the cost of a mild $6\%$ drop in accuracy for a global linear classifier trained on the embeddings.[2] In Appendix F.2, we additionally validate the SAE framework on MNIST (LeCun et al., 1998b), where we use $\Psi(x)$ directly rather than its reconstruction, as recovering an accurate $\hat{\Phi}(x)$ is much easier on this simpler dataset. We include training details in Appendix F.3.

**TTT baseline.** We define the neighborhood of a test point $x^\star$, denoted $\mathcal{D}_{x^\star}$, as its $k = 50$ nearest neighbors within the training set. Proximity is measured by the $L_2$-distance in a given feature space. For example, $\mathcal{D}_{x^\star}^\Psi$ denotes the neighborhood found in the space of CLIP embeddings. Since the CLIP embeddings are normalized, this is equivalent to using cosine similarity. The TTT procedure involves training a local linear classifier $W_{x^\star}$ on the neighborhood of $x^\star$:

$$W_{x^\star} := \underset{W \in \mathbb{R}^{1000 \times d_2}}{\arg\min} \frac{1}{k} \sum_{(x,y) \in \mathcal{D}_{x^\star}^{\hat{\Psi}}} \mathcal{L}(W\hat{\Psi}(x), y), \tag{1}$$

where $\mathcal{L}$ is the standard cross-entropy loss for the 1000 ImageNet classes. TTT in the estimated concept space is defined analogously using $\hat{\Phi}(x)$ and neighborhoods $\mathcal{D}_{x^\star}^{\hat{\Phi}}$.

## 3.2 RESULTS

**O1: The SAE preserves local geometry.** Our first hypothesis is that the SAE mapping preserves the angular relationships between a point and its neighbors. To test this, we select a neighborhood for a test point $x^\star$ in three different spaces: the original CLIP space ($\Psi$), the reconstructed space ($\hat{\Psi}$), and the estimated concept space ($\hat{\Phi}$). We then measure the average cosine similarity in the estimated concept space between $x^\star$ and points in each neighborhood. As shown in Figure 2, the distributions of cosine similarities are nearly identical regardless of the space used for neighbor selection. This suggests that the SAE projection to the concept space preserves the local geometric structure, supporting our first key observation.

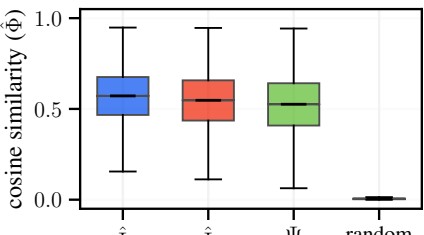

Figure 2: Average cosine similarity in the concept space ($\hat{\Phi}$) between a test point and its neighbors. Neighborhoods are selected in the original ($\Psi$), reconstructed ($\hat{\Psi}$), and concept ($\hat{\Phi}$) spaces.

**O2: Neighborhoods are supported by few concepts.** We hypothesize that the data within a local neighborhood can be explained by a small subset of concepts. To verify this, we train a TTT classifier for ImageNet on a masked version of the concept vectors, $\hat{\Phi}_m(x) = m \odot \hat{\Phi}(x)$, where $m \in \{0,1\}^{d_1}$ is a binary mask with $m_i = \mathbb{I}\{\theta_i > 0\}$ for some trainable parameter $\theta \in \mathbb{R}^{d_1}$. The

---

[2]Note that the SAE is trained in an unsupervised way, without explicitly retaining classification accuracy, yet using the SAE's features leads only to a minor drop in accuracy.

mask itself is learned for each neighborhood by optimizing the following objective with a straight-through estimator for the mask's gradients:[3]

$$W_{x^\star} := \underset{W,m}{\arg\min} \frac{1}{k} \sum_{(x,y) \in \mathcal{D}_{x^\star}^{\hat{\Phi}}} \mathcal{L}(W\hat{\Phi}_m(x), y) + \lambda \|m\|_2^2. \tag{2}$$

With a sparsity penalty of $\lambda = 0.2$, the learned masks are highly sparse, activating on average only $\|m\|_0 \approx 40$ concepts. This is substantially smaller than the total number of unique concepts active across the neighborhood, which is approximately 180. As shown in Table 1 (TTT column), this sparsely supported model performs on par with TTT on top of dense reconstructions $\hat{\Psi}(x)$[4]. This suggests that a small, adaptively chosen set of concepts is sufficient to capture the relevant information within a local region. We obtain similar results for the Gemma Scope SAE (Lieberum et al., 2024) on MNIST data, which we present in Appendix F.2.

|  | **Global** | **TTT** |
|---|---|---|
| $\hat{\Phi}(x)$ | $71.45 \pm 0.21$ | $72.64 \pm 0.20$ |
| $\hat{\Psi}(x)$ | $71.26 \pm 0.20$ | $72.56 \pm 0.19$ |

Table 1: ImageNet accuracy of globally trained linear models vs. TTT, with bootstrap standard errors.

Notably, a non-adaptive mask, such as one that only includes concepts active in the test point $x^\star$, performs poorly on ImageNet (71.51 %). The learned mask, in contrast, often excludes some of the test point's active concepts ($\|\hat{\Phi}(x^\star) \odot m\|_0 \approx 11 < 16 = s$), likely identifying and removing spurious features to improve generalization.

**O3: TTT in feature space implicitly finds a sparse solution.** While the adaptive masking in Equation (2) *explicitly* enforces a sparse solution, we find evidence that standard TTT in the feature space implicitly favors a solution that is sparse in the concept space. First, the TTT models trained on dense reconstructions $\hat{\Psi}(x)$ and sparse concepts $\hat{\Phi}_m(x)$ achieve nearly identical accuracy (cf. Table 1). Furthermore, their predictions agree in $\approx 89\%$ of cases, indicating that they learn functionally equivalent classifiers (apart from pathological examples). Figure 3 reinforces this by showing that both models lead to closely matched predictive distributions over the top-10 predicted classes. In Figure 3, we compare the ordered predicted probabilities for $\hat{\Phi}$ to the corresponding probabilities for $\hat{\Psi}$, matching the distributions' temperatures, and averaging over all test points. Their strong correspondence suggests that TTT on reconstructed embeddings is implicitly biased towards a sparse solution in the underlying concept space. This phenomenon may be linked to the implicit bias of optimization algorithms

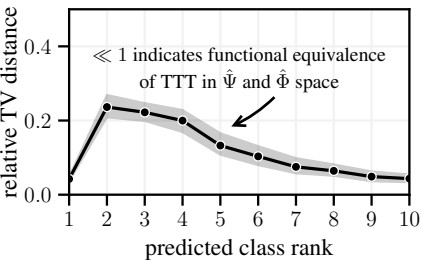

Figure 3: Comparison of predicted class probabilities for TTT models trained on dense reconstructions ($\hat{\Psi}$) and sparse concepts ($\hat{\Phi}$), relative to their magnitude. The small relative TV distance ($\ll 1$; defined in Appendix F.1) between both distributions indicates strong functional agreement between TTT in $\hat{\Psi}$ and $\hat{\Phi}$ space. We show 90 % bootstrap confidence intervals across 1000 test points.

(e.g., SGD or Adam), which are known to favor minimum-norm solutions (Gunasekar et al., 2018; Belkin et al., 2019; Frei et al., 2022). When the feature map superimposes concepts, this implicit bias may favor sparse solutions in the underlying concept space (Vaskevicius et al., 2019).

## 4 *WHEN* DOES SPECIALIZATION HELP?

After gaining some mechanistic understanding of TTT in Section 3, we next study *when* specialization through TTT improves over a globally trained model. In Section 5, we then relate our results from Sections 3-4, by providing theoretical support for *why* the LRH and our observations **O1-O3** may support our practically relevant findings from this section.

---

[3]We use the straight-through estimator $\nabla_\theta m = \text{sigmoid}(\theta/\tau)$ with $\tau = 0.1$.

[4]TTT in the concept space *without* extra masking achieves similar results. The fact that it also works with masking provides strong evidence that TTT adjusts only a few selected concepts in the neighborhood.

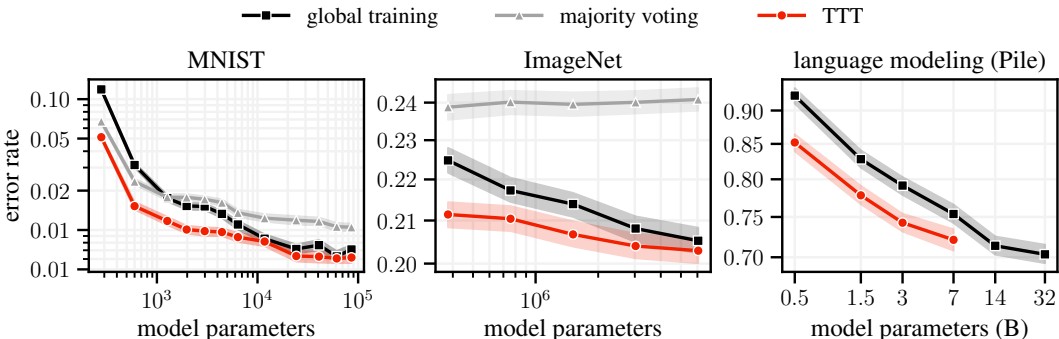

Figure 4: **Model scaling.** Error rates when scaling model size (classification error in image classification and bits per byte in language modeling). We evaluate a globally trained model (black) across different model sizes, as well as TTT (red) and a majority vote on the neighborhood (gray). While majority vote leads to a poor predictor with many classes (i.e., complex tasks), TTT consistently outperforms global training, with the performance gap shrinking as the model size increases. This supports our model's implication that TTT effectively recombines learned concepts, which is particularly beneficial when many concepts are superimposed in an underparameterized model. 7B is the largest model size we could train with LoRA on an NVIDIA RTX 4090. We do not evaluate majority voting for language modeling, since this has been shown to perform poorly by Hardt & Sun (2024).

## 4.1 SETTINGS

We consider the following three tasks, and provide implementation details in Appendix F:

- **MNIST.** We train a global classifier following the LeNet-5 (LeCun et al., 1998a) architecture with a cross-entropy loss on the MNIST (LeCun et al., 1998b) dataset of handwritten digits. To distinguish learning of concepts from TTT's ability to identify the correct weights for each locally relevant concept, we restrict TTT to updating only the last linear layer. This layer is fine-tuned to minimize the cross-entropy loss on the neighborhood of each test point (cf. Equation (1)), while all earlier layers remain fixed. As shown by additional experiments in Appendix D.2, end-to-end TTT does not yield a meaningful performance improvement, validating this design choice. We report classification error on the test set.

- **ImageNet.** We use a pre-trained CLIP ViT-B/32 vision transformer (Radford et al., 2021) to compute 512-dimensional embeddings of images. We then train a global linear classifier with a cross-entropy loss on the ImageNet-1K (Deng et al., 2009) dataset. As with MNIST, TTT updates only the last linear layer, and we report classification error on the test set.

- **Language modeling on the Pile.** To validate implications of our model on a real-world task, we consider language modeling on the Pile (Gao et al., 2020) dataset, restricting our use to data which is obtained and used in compliance with the terms of service of the data host. This version of the Pile contains 17 diverse and high-quality sub-datasets, including Wikipedia articles, academic papers, code, and more. We use the open-source implementation of Hardt & Sun (2024) which uses the Pile training set containing 210M sequences of total size 1.3TB for selecting neighbors, and evaluates on the Pile test set.[5] As recommended by Gao et al. (2020), we report *bits per byte*, which is proportional to the negative log-likelihood loss. As baseline for global training, we evaluate the Qwen2.5 family of base models (Qwen et al., 2025). Analogously to Hardt & Sun (2024), we implement TTT by fine-tuning a pre-trained LLM for a single gradient step each on 50 neighbors in the order that they are selected, from most similar to least similar. To perform TTT on a single GPU, we use LoRA (Hu et al., 2022), fine-tuning around 1 % of the model parameters.

Error bars correspond to 90 % confidence intervals computed via bootstrapping with 1000 samples.

---

[5]We split sequences to avoid retrieval-evaluation overlap (cf. Hardt & Sun (2024)), and evaluate on 1 % of the test set, corresponding to 1796 sequences.

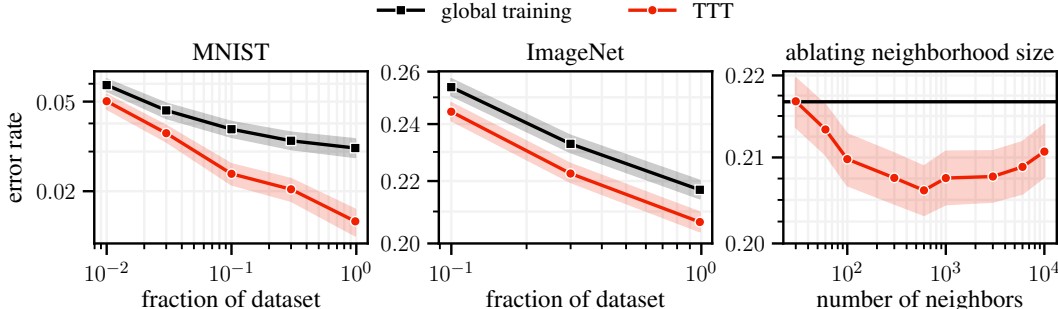

Figure 5: **Data scaling. Left & Middle:** Classification error rate of different models, trained on varying fractions of the MNIST and ImageNet training dataset. Notably on MNIST, we find that TTT learns more effectively from larger sample sizes than global training. **Right:** We vary the neighborhood size for TTT on ImageNet. We find that the optimal neighborhood size trades off statistical variance due to "too few examples" and "too many examples with irrelevant concepts".

## 4.2 RESULTS

**Scaling with model size.** We conduct a scaling study by varying the model size and comparing the performance of TTT against global training as well as a majority vote baseline over the neighborhood (i.e., a simple non-parametric approach). The results are shown in Figure 4. For MNIST, we train convolutional neural networks of varying sizes, as summarized in Appendix F. For ImageNet, we train multi-layer perceptrons on top of CLIP embeddings, varying the hidden dimension. In language modeling, we evaluate Qwen2.5 base models of sizes ranging from 0.5B to 32B parameters. We find across all tasks that TTT outperforms global training and majority vote, with the performance gap shrinking as the model size increases. We hypothesize that at a larger model size, fewer concepts have to be superimposed in latent space, leading to less interference when globally mapping latent representations to predictions. While a larger model size allows for better global disentanglement of concepts, TTT can compensate for limited model capacity by adapting the head to the specific concepts in a local neighborhood.

**Scaling with dataset size.** Next to performing a scaling study on model size, we also vary the dataset size. Figure 5 shows the results at a fixed model scale. We subsample the training datasets of ImageNet and MNIST to fractions ranging from 1 to 100 % of the original training set size, ensuring that all subsampled datasets are class-balanced. We then train global models and evaluate TTT on the respective test sets. We find that TTT consistently outperforms global training, with the slight trend of the performance gap widening as the dataset size increases. We hypothesize that larger datasets provide richer neighborhoods for local adaptation, enabling TTT to specialize more effectively to the specific concepts relevant to each test point.

**TTT improves predictions locally.** To validate that TTT improves predictions locally, we globally evaluate some TTT heads. As shown in Table 2, while improving accuracy on the test point and neighborhood, the global accuracy of these fixed TTT heads is significantly lower than that of the model trained on the entire dataset without TTT. These results confirm that, while TTT can provide localized performance improvements when adapted individually at test time, such benefits do not generalize when the same adaptation is applied globally. We further hypothesize that the neighborhood needs to be sufficiently large and diverse to span all relevant concepts. At the same time, the neighborhood needs to be sufficiently local to focus on *only* those concepts that are relevant to the test point. In Figure 5 (right), we support this hypothesis by varying the neighborhood size for TTT on ImageNet, and finding that the optimal neighborhood size trades off locality and diversity.

---

**Takeaway 1**

TTT locally improves predictions for underparameterized models, but its improvement diminishes as models become overparameterized.

---

|  | Global | TTT on Test Sample | TTT on Neighborhood | Global TTT |
|---|---|---|---|---|
| MNIST | $98.57 \pm 0.12$ | $99.01 \pm 0.10$ | $100.00 \pm 0.00$ | $36.38 \pm 0.16$ |
| ImageNet | $78.33 \pm 0.19$ | $79.39 \pm 0.18$ | $95.19 \pm 0.00$ | $77.04 \pm 0.06$ |

Table 2: Accuracy of a linear model trained on the full dataset (global), TTT evaluated on the test sample, TTT evaluated on the neighborhood, and of ten randomly selected local TTT heads evaluated on the entire test set (global TTT). We select ten TTT heads to keep the evaluation computationally tractable. The table reports bootstrap standard errors.

## 5 *Why* MAY SPECIALIZATION HELP?

Based on our mechanistic understanding gained in Section 3 through observations **O1-O3**, we next provide theoretical evidence that the LRH supports our practical observations from Section 4, therefore, potentially offering an understanding for *why* specialization is effective. To simplify the analysis, we consider a univariate regression setting with input space $\mathcal{X}$ and output space $\mathbb{R}$. Based on the LRH, we posit the existence of an $s$-sparse high-dimensional *concept space* $\Phi : \mathcal{X} \to \mathbb{R}^{d_1}$, and that the ground truth output is a linear combination of concepts, i.e., $\langle \Phi(x), w_\star \rangle$ for an arbitrary weight vector $w_\star \in \mathbb{R}^{d_1}$. Let us denote the learned *feature map* by $\Psi : \mathcal{X} \to \mathbb{R}^{d_2}$ and assume that the feature map is "underparameterized", i.e., $1 \leq d_2 \ll d_1$. We assume access to a training set $\mathcal{D}$ of size $N$.

### 5.1 ANALYZING THE IN-DISTRIBUTION TEST ERROR OF TTT

Informally, the LRH states that the model represents high-level concepts linearly as directions in some concept space, with the unembedding $w_\star$ defining the "meaning" of concepts. As is well-known from compressed sensing (Candes & Tao, 2006; Donoho, 2006), the feature map $\Psi$ can represent exponentially many concepts in *superposition* (Elhage et al., 2022), i.e., $d_1 \sim s \exp(d_2/s)$. However, as we will see in this section, even if the underparameterized feature map encodes the structure of the concept space, it is not straightforward to learn a linear mapping of these superimposed concepts. In contrast, we show in an idealized setting based on our empirical observations, that TTT can efficiently learn the meaning of exponentially many concepts from data, by *specializing* the model to the concepts relevant to the test data. We begin by restating our three key hypotheses based on observations from Section 3. We formalize these hypotheses in Appendix B and include proofs in Appendix C.

**Hypothesis 1: The feature space preserves the geometry of the concept space.** Based on **O1**, we posit that the learned feature map $\Psi$ preserves the similarity structure of the concept space $\Phi$. Let us denote by $\text{sim}_\Psi(x, x')$ a similarity measure in $\Psi$-space,[6] which defines a neighborhood:

**Definition 1** (Neighborhood). *For a test point $x^\star \in \mathcal{X}$ and a radius $r \geq 0$, the neighborhood of size $k$ in the training set is defined based on the similarity measure in terms of learned features:*

$$B_{x^\star}^\Psi(r) := \{(x, y) \in \mathcal{D} \mid \text{sim}_\Psi(x, x^\star) \geq 1 - r\}, \quad k := |B_{x^\star}^\Psi(r)|.$$

We assume that the neighborhood in feature space is contained within a slightly larger neighborhood in concept space, i.e., $B_{x^\star}^\Psi(r) \subseteq B_{x^\star}^\Phi(r + \delta)$. For example, by the classical Johnson-Lindenstrauss lemma (Johnson & Lindenstrauss, 1984; Vershynin, 2018) this assumption is satisfied with high probability for $\delta \leq O(\sqrt{\log(N)/d_2})$ if the lower-dimensional feature map $\Psi(x)$ is a random projection of $\Phi(x)$ and $\text{sim}$ measures angles.

**Hypothesis 2: Neighborhoods are supported by few concepts.** In Section 3, we have made the surprising observation that the neighborhood of a test point $x^\star$ is explained by only a few active concepts (cf. **O2**). Based on this, we assume that locally there exists an $\Theta(s)$-sparse concept vector $w_{x^*} \in \mathbb{R}^{d_1}$ that approximates $\langle \Phi(x), w_\star \rangle$ over the test point's neighborhood. This assumption may seem unrealistic at first, since the neighborhood of $x^\star$ has $k$ samples and up to $s \cdot k$ active concepts. Yet, as we observe empirically in Section 3, TTT learns a $\Theta(s)$-sparse regressor that does well on the entire neighborhood if the neighborhood size is sufficiently small.

---

[6]sim may be any measure such that neighborhoods in feature and concept space approximately coincide.

**Hypothesis 3: TTT implicitly regularizes towards sparsity in concept space.** Based on **O3**, we hypothesize that TTT finds sparse solutions in concept space, even without explicit regularization. Concretely, we assume that TTT in $\Psi$-space implicitly finds a $\Theta(s)$-sparse solution when mapped to concept space.

With these hypotheses, we can bound the in-distribution test error of TTT using techniques from sparse recovery (Bickel et al., 2009; Van De Geer & Bühlmann, 2009):[7]

**Proposition** (informal, see Proposition 4 in the appendix). *Let $\Phi : \mathcal{X} \to \mathbb{R}^{d_1}$ be an s-sparse concept space and $\Psi : \mathcal{X} \to \mathbb{R}^{d_2}$ be a learned feature map with $d_2 \ll d_1$. Let $f(x) = \langle \Phi(x), w_\star \rangle$ be the ground truth function and labels be $\sigma^2$-subgaussian. Let $x^\star$ be a test point with a neighborhood (in $\Psi$-space) of sufficiently small size $k$ such that Hypotheses 1–3 hold. Let the learned features $\Psi$ be sufficiently expressive to represent $f$ locally (in particular, $d_2 \geq \Omega(s \log d_1)$). We denote by $\hat{v}_{x^\star}^{TTT}$ the local empirical risk minimizer on the neighborhood of $x^\star$.*

*Then, under standard regularity conditions for sparse recovery and with high probability over the sampling of the data,*

$$(f(x^\star) - \langle \Psi(x^\star), \hat{v}_{x^\star}^{TTT} \rangle)^2 \leq O\left( \frac{\sigma^2 s \log(d_1/s)}{k} \right).$$

*This is the standard minimax optimal rate from sparse recovery (Raskutti et al., 2011, Theorem 1).*

> **Takeaway 2**
>
> In this idealized model, TTT can locally learn a function from very few samples that activate similar concepts as the test point, even when the feature map is underparameterized.

We compare this error bound to the generalization error of training a global model $\hat{v}^{\text{global}}$ on all data:

**Proposition** (informal, see §C.3 in the appendix). *We construct an instance of our model with $f(x) = 1$ where $\Psi$ is a random projection of $\Phi$. In this instance, the error of the global model, when averaged over random realizations of the feature map $\Psi$, is $\mathbb{E}_\Psi[(f(x) - \langle \Psi(x), \hat{v}^{global} \rangle)^2] = 1 - \frac{d_2}{d_1}$.*

As one would expect, if the global model is not underparameterized, i.e., $d_2 = d_1$, the error of global training is zero. On the other hand, the trivial global model $\hat{v}^{\text{global}} = \mathbf{0}$ has error 1. As the model size $d_2$ shrinks, the error of global training increases towards 1. As the number of distinct concepts $d_1$ increases, the error of global training also increases, approaching 1 as $d_1 \to \infty$.

> **Takeaway 3**
>
> The example illustrates that when concepts are superimposed in an underparameterized feature space, a linear head cannot globally disentangle the meaning of all concepts.

We expand on our theoretical results in the appendix as follows: First, we explore whether one can understand TTT under the LRH through the lens of statistical learning theory. Specifically, in Appendix A.1, we explore this direction using notions from low-degree polynomials and hypercontractivity (Klivans et al., 2008; Paouris et al., 2022; Damian et al., 2024; Bizeul & Klartag, 2025, and references therein). Additionally, in Appendix A.2, we contrast TTT to classical non-parametric methods (Fix & Hodges Jr., 1951; Nadaraya, 1964; Watson, 1964) such as majority voting, which underperform in our experiments (cf. Section 4).

## 6 CONCLUSION

This work introduces a framework, supported by new empirical findings, for understanding the effectiveness of TTT on in-distribution data, based on the hypothesis that foundation models are globally underparameterized. We hypothesize that TTT facilitates *specialization after generalization*, temporarily reallocating model capacity to concepts relevant to the immediate test task. We formalize

---

[7]Our results indicate that TTT may not converge quickly (or at all) if the label noise $\sigma^2$ is large or any of Hypotheses 1-3 were to be violated. Notably, the neighborhood size $k$ needs to be sufficiently small for Hypothesis 2 to hold.

this intuition under the linear representation hypothesis, and show how TTT can efficiently recover the local meaning of superimposed concepts (§5). Our trained sparse autoencoders reveal that local neighborhoods are indeed supported by few concepts and that TTT implicitly favors sparse solutions in the concept space (§3). Finally, scaling studies across vision and language tasks confirm that TTT yields the largest gains in the underparameterized regime (§4).

A better understanding of specialization in foundation models opens up several exciting directions for future research. An interesting question is understanding what determines the optimal neighborhood size and whether it depends on the test point. Furthermore, it would be interesting to analyze the compute-efficiency trade-offs of TTT; estimating at which model scale and inference budget TTT becomes beneficial. The importance of specialization is also evident in online learning, where increasingly specialized experience may require ever-larger models to fit globally; TTT offers a potential solution by enabling local adaptation. Particularly interesting would therefore be extending this framework beyond supervised learning to test-time online reinforcement learning, which has recently shown empirical success.

## ACKNOWLEDGMENTS

We would like to thank Reese Pathak and Pierre Bizeul for helpful discussions. We also thank Bruce Lee, Celestine Mendler-Dünner, and Lars Lorch for feedback on early versions of the paper. JH was supported by the Swiss National Science Foundation under NCCR Automation, grant agreement 51NF40 180545. PW was supported by the Max Planck ETH Center for Learning Systems. AS was supported by the Swiss National Science Foundation under grant 204439. GK conducted the initial part of this work during his visit to the IDEAL Institute, hosted by Lev Reyzin, which was supported by NSF ECCS-2217023.

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

# APPENDICES

## CONTENTS

## A  DISCUSSION

### A.1  LIMITATIONS OF GLOBAL LEARNING UNDER THE LRH

Next, we demonstrate that under the LRH, obtaining a global classifier requires a quasi-polynomial number of samples *in the dimension of the sparse concept space* when analyzed within the "low-degree polynomial" framework, commonly used in learning theory (Kalai et al., 2008; Klivans et al., 2024; Damian et al., 2022; 2024, and references therein). In a nutshell, the idea is to think about the behavior of underparametrization as the behavior of an approximating low-degree polynomial.

We consider the Gaussian measure $\gamma_d$ on $\mathbb{R}^d$ and introduce the family of $s$-sparse, $k$-locally linear functions (with $k$ "cells"), denoted by $\mathcal{F}_{d,s,k} \subset \{\mathbb{R}^d \to \mathbb{R}\}$, defined as

$$\mathcal{F}_{d,s,k} := \left\{ f^\star(x) = \sum_{i=1}^{k} w_i^\top x \cdot \mathbf{1}_{K_i} : \forall 1 \leq i < j \leq k \; \gamma_d(K_i \cap K_j) = 0, \|w_i\|_0 \leq s, \right.$$

$$\left. \|w_i\|_2 \lesssim 1, \|f^\star\|_{L_2(\gamma_d)} \asymp 1, \sum_{i=1}^{d} \|\partial_i f^\star\|_{L_1(\gamma_d)}^2 \lesssim \frac{s}{d} \right\}. \tag{3}$$

The final condition defines the sparsity index derived from the $L_1$-$L_2$ Talagrand's inequality (cf. the monograph of Chatterjee (2014) and the survey of Ledoux (2019)), and note that the cells $K_1, \ldots, K_k$ may depend on the function $f^\star$. For intuition, consider the function $f^\star(x) = \|x\|_\infty$; here $k = d$, $s = 1$, and the sparsity index condition holds with $\sum_{i=1}^{d} \|\partial_i f^\star\|_{L_1(\gamma_d)}^2 \lesssim 1/d$.

We argue that when $s \asymp 1$ and $k$ is polynomial in $d$, the functions in $\mathcal{F}_{d,s,k}$ predominantly lie in the "high" frequencies of the Hermite polynomial basis. Specifically, by leveraging results of Paouris et al. (2022), we show that for some constants $c, C > 0$, it holds that

$$\left\| \mathbb{E}[f^\star] + \sum_{c \log(d) \leq m \leq C \log(d)} \mathcal{P}_m(f^\star) \right\|_{L_2(\gamma_d)}^2 \in (0.1, 0.9), \tag{4}$$

where $\mathcal{P}_m(\cdot)$ denotes the orthogonal projection onto the $m$-degree Hermite polynomial basis. The main result of this part is the following:

**Proposition 2** (informal, see Proposition 11 below). *Assume that $\sigma \lesssim 1$, $k \asymp \mathrm{Poly}(d)$, and $s \asymp 1$. Then Equation (4) holds. Furthermore, for $n \gtrsim \exp(\Omega((\log d)^2))$, there exists a polynomial-time algorithm $\mathcal{A}$ such that*

$$\sup_{f^\star \in \mathcal{F}_{d,s,k}} \mathbb{E}_{\mathcal{D}} \|\mathcal{A}(\mathcal{D}) - f^\star\|_{L_2(\gamma_d)} \leq 0.1,$$

*and under the low-degree polynomial conjecture, this bound is sharp for few classes of algorithms.*

We refer to the survey of Reyzin (2020) for more details on the low-degree polynomial conjecture. We emphasize that our assumptions on $\mathcal{F}_{d,s,k}$ are much less restrictive than imposing a uniform bounded Lipschitz constraint (or an average $L_2$ Lipschitz constraint), as the Lipschitz constant may be high on the boundary between two cells, and the results of Bizeul & Klartag (2025) show that Lipschitz functions can be learned with a polynomial number of samples.

Roughly speaking, from a geometric view, this spectral concentration implies that most of the energy of the coefficients *is localized at the decision boundaries* between the cells $\{K_i\}$. An "underparameterized" global classifier, in this case, the best low-degree approximation, necessarily smooths these boundaries. Therefore, by isoperimetry in high dimensions, this smoothing leads to significant overlap between the decision boundaries of cells. Since most samples fall in the areas of these "distorted/smoothed" decision boundaries, we need many samples to learn such functions.[8] The latter aligns with previous observations regarding the required complexity (and the lack of robustness) of foundation models (e.g., Bubeck & Sellke, 2021).

## A.2 TTT vs. NON-PARAMETRIC METHODS

While TTT utilizes local neighborhoods, superficially resembling majority voting (i.e., $k$-NN; Fix & Hodges Jr., 1951) or kernel regression (Nadaraya, 1964; Watson, 1964), its mechanism is fundamentally different. Non-parametric methods generally require the target function to be locally smooth or constant in the feature space ($\Psi$). When this assumption fails in high dimensions, their performance degrades rapidly—the so-called curse of dimensionality (e.g., Hastie et al., 2009; Stone, 1982). Our framework explains this failure mode under the LRH. The superposition of concepts into the underparameterized feature space leads to a function that is locally complex and non-smooth within $\Psi$, even though it is simple (sparse linear) in $\Phi$. This leads to ambiguous neighborhoods in $\Psi$ where

---

[8]It is well-known that *even* for 1-Lipschitz function, its best low-degree polynomial (it terms of $L_2$), is highly non-Lipschitz, cf. Bizeul & Klartag (2025).

samples share concepts but possess different labels. Simple averaging (like $k$-NN) cannot resolve this ambiguity as it relies on a local smoothness that may not hold in $\Psi$.

In contrast, TTT performs local specialization by optimizing a local *parametric* model. TTT exploits the underlying sparse structure in the concept space $\Phi$ (§3), effectively executing sparse recovery (§5.1). This allows TTT to disentangle the local meaning of superimposed features, making it substantially more effective than majority voting, as confirmed by our experiments (§4).

# B   FORMAL ASSUMPTIONS

Our key assumption is that for any given input, only a few concepts are active:

**Assumption 1** (**linear representation hypothesis** (sparse concept space))**.** *For all $x \in \mathcal{X}$, the concept vector $\Phi(x)$ is $s$-sparse, i.e., $\|\Phi(x)\|_0 \leq s$.*

Next, we formalize the series of hypotheses from Section 5.1.

**Hypothesis 1: Feature space preserves the geometry of the concept space.**   We hypothesize that the learned feature map $\Psi$ preserves the similarity structure of the concept space $\Phi$. Let us denote by $\operatorname{sim}_\Psi(x, x')$ a similarity measure in $\Psi$-space such as cosine similarity.

**Assumption 2** (Neighborhood preservation)**.** *The learned feature map $\Psi$ preserves the similarity structure of the concept map $\Phi$. There exists a distortion $\eta_{ang} \geq 0$ such that $B_{x^\star}^\Psi(r)$ is contained within the concept neighborhood $B_{x^\star}^\Phi(r + \eta_{ang})$.*

**Hypothesis 2: Neighborhoods are supported by few concepts.**   Experimentally, we make the additional surprising observation that the neighborhood of a test point $x^\star$ is explained by only a few active concepts. Let us denote the corresponding feature matrices as $\mathbf{\Phi}_{x^\star} \in \mathbb{R}^{k \times d_1}$ and $\mathbf{\Psi}_{x^\star} \in \mathbb{R}^{k \times d_2}$, and the observation vector by $\boldsymbol{y}_{x^\star} \in \mathbb{R}^k$. We assume:

**Assumption 3** (Local simplicity)**.** *Locally, the ground truth function is well-approximated by a sparse model. There exists an $s'$-sparse concept vector $w_{x^*} \in \mathbb{R}^{d_1}$ with $s' \in \Theta(s)$ such that the average approximation error over the neighborhood is bounded:*

$$\frac{1}{k}\|\langle \mathbf{\Phi}_{x^\star}, w_\star - w_{x^\star}\rangle\|_2^2 \leq \eta_{spa}(r).$$

**Learned features need to be sufficiently expressive.**   Next, we quantify the expressivity of the learned features $\Psi$. Naturally, features need to be sufficiently expressive for *any* linear model in feature space to approximate the ground truth function. First, we make the relatively straightforward assumption that *locally*, the learned features are a linear recombination of concepts.

**Assumption 4** (Local linearity of learned features)**.** *Locally, the learned features are a linear recombination of concepts. That is, there exists some $P_{x^\star} \in \mathbb{R}^{d_2 \times d_1}$ such that $\Psi(x) = P_{x^\star}\Phi(x)$ for all $x \in \{x^\star\} \cup B_{x^\star}^\Psi(r)$.*

Even if the global map from concepts $\Phi(x)$ to features $\Psi(x)$ may be non-linear, A4 posits that a local linear approximation is sufficient within a small neighborhood, where the local behavior can often be approximated by a first-order Taylor expansion. This assumption is further supported by the effectiveness of linear decoders in sparse autoencoders (Elhage et al., 2022).

**Assumption 5** (Expressivity of learned features)**.** *The feature map $\Psi$ is expressive enough to represent the local function $w_{x^\star}$. We consider the set of $s'$-sparse weight vectors in $\Phi$-space that are linearly representable by the features $\Psi$. By local linearity (A4), this means a vector $w$ must lie in the image of $P_{x^\star}^\top$ (i.e., $w = P_{x^\star}^\top v$ for some $v \in \mathbb{R}^{d_2}$). Let $\tilde{w}_{x^\star}$ be the vector in this set that best approximates the predictions of $w_{x^\star}$ over the neighborhood. We assume that the resulting representation error is bounded:*

$$\frac{1}{k}\|\langle \mathbf{\Phi}_{x^\star}, w_{x^\star} - \tilde{w}_{x^\star}\rangle\|_2^2 \leq \eta_{rep}.$$

If $w_{x^\star}$ is not in the image of $P_{x^\star}^\top$, there is no corresponding vector $v$ in the feature space that can replicate its behavior. This means the compression defined by $P_{x^\star}$ has discarded information necessary to represent $w_{x^\star}$. Thus, A5 highlights the importance of pre-training: for the representation error $\eta_{\text{rep}}$ to be small, the feature map needs to learn sufficient structure to represent the ground truth function locally.

**Hypothesis 3: TTT implicitly regularizes towards sparsity in concept space.** We find experimentally (§3) that TTT solutions often exhibit behavior consistent with sparsity in the concept space, even without explicit regularization. To facilitate theoretical analysis using sparse recovery frameworks, we analyze an idealized TTT estimator that explicitly enforces this observed sparsity:

**Assumption 6** (Implicit regularization). *We assume that TTT is implicitly regularized towards solutions which are sparse in concept space:*

$$\hat{v}_{x^\star}^{TTT} = \arg\min_{v \in \mathbb{R}^{d_2}} \frac{1}{k} \|\boldsymbol{\Psi}_{x^\star} v - \boldsymbol{y}_{x^\star}\|_2^2 \quad \textit{subject to } \|P_{x^\star}^\top v\|_0 \leq s'. \tag{5}$$

While standard TTT omits the explicit constraint, we use this formulation to analyze the specialization mechanism that we observe empirically. It remains to state standard assumptions:

**Assumption 7** (Bounded concepts). *Concepts are bounded in $L_\infty$ and $L_2$ norm, i.e., for all $x \in \mathcal{X}$, $\|\Phi(x)\|_\infty \leq C_{\Phi,\infty}$ and $\|\Phi(x)\|_2 \leq C_{\Phi,2}$.*

**Assumption 8** (Linear model with homoscedastic noise). *Data follows a linear model in the concept space: $y = \langle \Phi(x), w_\star \rangle + \varepsilon$. The noise $\varepsilon$ is i.i.d. zero-mean and $\sigma^2$-subgaussian.*

Finally, we assume that $\boldsymbol{\Psi}_{x^\star}$ satisfies the generalized restricted eigenvalue (GRE) condition. Combined with local linearity (A4), the GRE is simply the standard restricted eigenvalue condition on the local concept design matrix $\boldsymbol{\Phi}_{x^\star}$, a condition that is fundamental to guarantee stable recovery in sparse regression (Bickel et al., 2009; Van De Geer & Bühlmann, 2009).

**Assumption 9** (Generalized restricted eigenvalue (GRE) condition). *The local design matrix $\boldsymbol{\Psi}_{x^\star}$ satisfies the GRE at order $2s'$ with respect to $P_{x^\star}$. There exists $\kappa > 0$ such that for all $v$ with $\|P_{x^\star}^\top v\|_0 \leq 2s'$:*

$$\frac{1}{k} \|\boldsymbol{\Psi}_{x^\star} v\|_2^2 \geq \kappa \cdot \|P_{x^\star}^\top v\|_2^2. \tag{6}$$

With this, we are ready to state our result of this section.

**Proposition 3** (Informal version of Proposition 4). *Fix any test point $x^\star \in \mathcal{X}$ and any $\delta \in (0,1)$. Let A1–A9 hold. Define the inherent misspecification error of any sparse linear approximation as $\eta_{inherent} := \langle \Phi(x^\star), w_\star - \tilde{w}_{x^\star} \rangle^2$ and the misspecification error on the neighborhood as $\eta_{mis} := \eta_{spa}(r + \eta_{ang}) + \eta_{rep}$.*

*Then, with probability $1 - \delta$ over the sampling of the data,*

$$\mathbb{E}\left[ \left( y^\star - \langle \Psi(x^\star), \hat{v}_{x^\star}^{TTT} \rangle \right)^2 \right] \leq \sigma^2 + O\left( \frac{\sigma^2 s \log(d_1/s)}{k} \right) + O(s\, \eta_{mis}) + \eta_{inherent}.$$

We prove the result in Appendix C and briefly highlight several aspects of the error bound:

1. The fast rate $\widetilde{O}(s/k)$ explains why TTT can learn from few samples.
2. The optimal neighborhood size $k$ depends on the tradeoff between variance (i.e., $\widetilde{O}(s/k)$) and bias (from $\eta_{spa}$). Note that A3 assumes the neighborhood is described by a $\Theta(s)$-sparse model, which is only possible if $k \ll N$ and the neighborhood is sufficiently "local".
3. The error grows linearly with more active concepts $s$ and is only logarithmically dependent on the total number of concepts $d_1$. In contrast, a larger feature dimension $d_2$ can reduce misspecification error through $\eta_{ang}$ and $\eta_{rep}$.

## C PROOFS

### C.1 PROOF OF PROPOSITION 3

We begin by stating the formal version of Proposition 3:

**Proposition 4.** *Fix any test point $x^\star \in \mathcal{X}$ and any $\delta \in (0,1)$. Let A1–A9 hold. Let $k$ be the neighborhood size and $s'$ the sparsity in concept space. Let concept vectors be bounded in $L_\infty$ and $L_2$ space with constants $C_{\Phi,\infty}, C_{\Phi,2} > 0$. Assume the GRE holds with $\kappa > 0$. Define the inherent misspecification bias of any sparse linear approximation as $\mathcal{E}_{inherent}^2 := \langle \Phi(x^\star), w_\star - \tilde{w}_{x^\star} \rangle^2$.*

*With probability $1 - \delta$ (over sampling of the data), the squared prediction error is bounded as:*

$$\mathbb{E}\left[\left(y^\star - \langle \Psi(x^\star), \hat{v}_{x^\star}^{TTT}\rangle\right)^2\right] \leq \sigma^2 + \left(\mathcal{E}_{inherent} + \mathcal{E}_{estimation}\right)^2$$

*where the squared estimation error $\mathcal{E}_{estimation}^2$ achieves rate $\widetilde{O}(s/k)$ with $C$ a universal constant:*

$$\mathcal{E}_{estimation}^2 \leq \frac{CC_{\Phi,2}^2 C_{\Phi,\infty}^2}{\kappa^2} \cdot s' \cdot \left(\sigma^2 \frac{\log(d_1/s'\delta)}{k} + \underbrace{\eta_{spa}(r + \eta_{ang}) + \eta_{rep}}_{misspecification}\right). \tag{7}$$

*Proof of Proposition 4.* **Step 1: Data decomposition and comparison to oracle.** By A5, there exists some "oracle" $\tilde{v}_{x^\star} \in \mathbb{R}^{d_2}$ such that $P_{x^\star}^\top \tilde{v}_{x^\star} = \tilde{w}_{x^\star}$.

We decompose the observations $\boldsymbol{y}_{x^\star}$ (cf. A8).

$$\begin{aligned}
\boldsymbol{y}_{x^\star} &= \boldsymbol{\Phi}_{x^\star} w_\star + \varepsilon \\
&= \boldsymbol{\Phi}_{x^\star}\tilde{w}_{x^\star} + \underbrace{(\boldsymbol{\Phi}_{x^\star}w_\star - \boldsymbol{\Phi}_{x^\star}w_{x^\star}) + (\boldsymbol{\Phi}_{x^\star}w_{x^\star} - \boldsymbol{\Phi}_{x^\star}\tilde{w}_{x^\star})}_{\Delta \text{ (total misspecification)}} + \varepsilon.
\end{aligned}$$

Note that using A2, A3, A5, the average squared magnitude of $\Delta$ is bounded by

$$\frac{1}{k}\|\Delta\|_2^2 \leq \eta'(r,s,d_2) := 2(\eta_{\text{spa}}(r + \eta_{\text{ang}}, s) + \eta_{\text{rep}}(d_2)).$$

Let $h = \tilde{v}_{x^\star} - \hat{v}_{x^\star}^{\text{TTT}}$. Since by A5, both $\tilde{v}_{x^\star}$ and $\hat{v}_{x^\star}^{\text{TTT}}$ satisfy the constraint of Equation (5), the error vector $h$ is sparse in the concept space: $\|P_{x^\star}^\top h\|_0 \leq 2s'$.

**Step 2:** By the optimality of $\hat{v}_{x^\star}^{\text{TTT}}$ (cf. A6):

$$\frac{1}{k}\|\boldsymbol{\Psi}_{x^\star}\hat{v}_{x^\star}^{\text{TTT}} - \boldsymbol{y}_{x^\star}\|_2^2 \leq \frac{1}{k}\|\boldsymbol{\Psi}_{x^\star}\tilde{v}_{x^\star} - \boldsymbol{y}_{x^\star}\|_2^2.$$

Substituting $\boldsymbol{y}_{x^\star} = \boldsymbol{\Psi}_{x^\star}\tilde{v}_{x^\star} + \Delta + \varepsilon$ (using local linearity, A4, $\boldsymbol{\Psi}_{x^\star}\tilde{v}_{x^\star} = \boldsymbol{\Phi}_{x^\star}\tilde{w}_{x^\star}$):

$$\frac{1}{k}\| - \boldsymbol{\Psi}_{x^\star}h - (\Delta + \varepsilon)\|_2^2 \leq \frac{1}{k}\|\Delta + \varepsilon\|_2^2.$$

Expanding the left hand side:

$$\frac{1}{k}(\|\boldsymbol{\Psi}_{x^\star}h\|_2^2 + 2h^\top\boldsymbol{\Psi}_{x^\star}^\top(\Delta + \varepsilon) + \|\Delta + \varepsilon\|_2^2) \leq \frac{1}{k}\|\Delta + \varepsilon\|_2^2.$$

Rearranging yields the basic inequality:

$$\frac{1}{k}\|\boldsymbol{\Psi}_{x^\star}h\|_2^2 \leq -\frac{2}{k}h^\top\boldsymbol{\Psi}_{x^\star}^\top(\Delta + \varepsilon). \tag{8}$$

**Step 3:** We analyze the right hand side of Equation (8). Using local linearity (cf. A4), $h^\top\boldsymbol{\Psi}_{x^\star}^\top = (P_{x^\star}^\top h)^\top\boldsymbol{\Phi}_{x^\star}^\top$. Let $Z = \frac{1}{k}\boldsymbol{\Phi}_{x^\star}^\top(\Delta + \varepsilon)$ be the total concept score. Then,

$$\frac{1}{k}h^\top\boldsymbol{\Psi}_{x^\star}^\top(\Delta + \varepsilon) = \langle P_{x^\star}^\top h, Z\rangle.$$

We bound the inner product using the sparse dual norm, since $\|P_{x^\star}^\top h\|_0 \leq 2s'$:

$$|\langle P_{x^\star}^\top h, Z\rangle| \leq \|P_{x^\star}^\top h\|_2 \cdot \|Z\|_{2,2s'}^*.$$

We condition on the high probability event (w.p. $1 - \delta$) of Lemma 6 and use Lemma 7 (with $\eta = \eta'(r,s,d_2)$).

$$\|Z\|_{2,2s'}^* \leq \Lambda + \eta_\Delta := \Gamma.$$

Substituting this back into Equation (8) gives:

$$\frac{1}{k}\|\boldsymbol{\Psi}_{x^\star}h\|_2^2 \leq 2\Gamma\|P_{x^\star}^\top h\|_2. \tag{9}$$

**Step 4: Applying GRE.** Since $\|P_{x^\star}^\top h\|_0 \leq 2s'$, we can apply the GRE (cf. A9) to Equation (9):

$$\kappa \|P_{x^\star}^\top h\|_2^2 \leq 2\Gamma \|P_{x^\star}^\top h\|_2.$$

This yields a bound on the $L_2$ estimation error:

$$\|P_{x^\star}^\top h\|_2 \leq \frac{2\Gamma}{\kappa}. \tag{10}$$

**Step 5: Bounding the prediction error.** We next decompose the total prediction error,

$$
\begin{aligned}
\mathcal{E} &= (y^\star - \langle \Psi(x^\star), \hat{v}_{x^\star}^{\text{TTT}} \rangle) \\
&= (y^\star - \langle \Psi(x^\star), \tilde{v}_{x^\star} \rangle) + \langle \Psi(x^\star), \tilde{v}_{x^\star} - \hat{v}_{x^\star}^{\text{TTT}} \rangle \\
&= (y^\star - \langle \Psi(x^\star), \tilde{v}_{x^\star} \rangle) + \langle \Psi(x^\star), h \rangle \\
&= (y^\star - \langle \Phi(x^\star), \tilde{w}_{x^\star} \rangle) + \langle \Phi(x^\star), P_{x^\star}^\top h \rangle && \text{(local linearity, A4)} \\
&= \varepsilon + \underbrace{\langle \Phi(x^\star), w_\star - \tilde{w}_{x^\star} \rangle}_{\mathcal{E}_{\text{inherent}}} + \underbrace{\langle \Phi(x^\star), P_{x^\star}^\top h \rangle}_{\mathcal{E}_{\text{estimation}}}. && \text{(data model, A8)}
\end{aligned}
$$

We next bound the estimation error $\mathcal{E}_{\text{estimation}}$ using A7 ($L_2$ bound):

$$|\mathcal{E}_{\text{estimation}}| = |\langle \Phi(x^\star), P_{x^\star}^\top h \rangle| \leq \|\Phi(x^\star)\|_2 \|P_{x^\star}^\top h\|_2 \leq C_{\Phi,2} \|P_{x^\star}^\top h\|_2.$$

Combining this with Equation (10) gives:

$$|\mathcal{E}_{\text{estimation}}| \leq \frac{2C_{\Phi,2}\Gamma}{\kappa}.$$

Hence, the expected squared error is $\mathbb{E}[\mathcal{E}^2] \leq \sigma^2 + (|\mathcal{E}_{\text{inherent}}| + |\mathcal{E}_{\text{estimation}}|)^2$.

**Step 6: Finalizing the bound.** Finally, we resolve the dependencies and compute $\mathcal{E}_{\text{estimation}}^2$. Using $(a+b)^2 \leq 2a^2 + 2b^2$:

$$\mathcal{E}_{\text{estimation}}^2 \leq \frac{4C_{\Phi,2}^2}{\kappa^2}\Gamma^2 \leq \frac{8C_{\Phi,2}^2}{\kappa^2}(\Lambda^2 + \eta_\Delta^2).$$

Substituting the definitions from Lemmas 6 and 7:

$$
\begin{aligned}
\Lambda^2 &= C_H^2 C_{\Phi,\infty}^2 \sigma^2 \frac{s'}{k}\Big( \log(d_1/s') + \log(1/\delta) \Big), \\
\eta_\Delta^2 &= 2s' C_{\Phi,\infty}^2 \eta'(r, s, d_2).
\end{aligned}
$$

Therefore,

$$\mathcal{E}_{\text{estimation}}^2 \leq \frac{8C_{\Phi,2}^2 C_{\Phi,\infty}^2}{\kappa^2} \cdot s' \cdot \left( C_H^2 \sigma^2 \frac{\log(d_1/s'\delta)}{k} + 4\eta'(r, s, d_2) \right).$$

Combining the universal constants into $C$ yields the final result. $\qquad \square$

## C.2 Concentration Bounds

We utilize the sparse dual norm to analyze the correlation between sparse vectors and the noise / misspecification.

**Definition 5** (Sparse dual norm). *We define the sparse $L_2$ dual norm of a vector $z \in \mathbb{R}^{d_1}$ as:*

$$\|z\|_{2,m}^* := \sup_{\|u\|_0 \leq m, \|u\|_2=1} \langle u, z \rangle = \max_{S:|S|=m} \|z_S\|_2.$$

**Lemma 6** (Sparse noise concentration). *Under A7 ($L_\infty$ bound) and A8 (subgaussian noise), there exists a universal constant $C_H > 0$ such that for any $\delta \in (0,1)$, with probability at least $1-\delta$:*

$$\left\| \frac{1}{k}\mathbf{\Phi}_{x^\star}^\top \varepsilon \right\|_{2,2s'}^* \leq \Lambda := C_H \cdot C_{\Phi,\infty} \cdot \sigma \sqrt{\frac{s'}{k}\Big( \log(d_1/s') + \log(1/\delta) \Big)}.$$

*Proof.* Let $Z_\varepsilon = \frac{1}{k}\boldsymbol{\Phi}_{x^\star}^\top\varepsilon$ and $m = 2s'$. The sparse dual norm is the supremum of a stochastic process indexed by the set of sparse unit vectors, $\mathcal{U}_m = \{u \in \mathbb{R}^{d_1} : \|u\|_0 \leq m, \|u\|_2 = 1\}$.

$$\|Z_\varepsilon\|_{2,m}^* = \sup_{u \in \mathcal{U}_m} \langle u, Z_\varepsilon \rangle.$$

For any $u \in \mathcal{U}_m$, the random variable $X_u = \langle u, Z_\varepsilon \rangle = \frac{1}{k}\sum_{i=1}^k \varepsilon_i \langle \Phi(x_i), u \rangle$ is a sum of independent centered subgaussian variables. Its variance is uniformly bounded:

$$
\begin{aligned}
\mathrm{Var}(X_u) &= \frac{1}{k^2}\sum_{i=1}^k \mathrm{Var}(\varepsilon_i)\,\langle \Phi(x_i), u \rangle^2 \\
&\leq \frac{\sigma^2}{k^2}\sum_{i=1}^k \langle \Phi(x_i), u \rangle^2 \leq \frac{\sigma^2}{k^2}\sum_{i=1}^k \left(\|\Phi(x_i)\|_\infty \|u\|_1\right)^2 \\
&\leq \frac{\sigma^2}{k^2}\sum_{i=1}^k \left(C_{\Phi,\infty}\sqrt{m}\|u\|_2\right)^2 \leq \frac{\sigma^2}{k^2}\left(k \cdot m C_{\Phi,\infty}^2\right) = \frac{m\sigma^2 C_{\Phi,\infty}^2}{k}.
\end{aligned}
$$

Moreover, the process has subgaussian increments: for any $u, v \in \mathcal{U}_m$,

$$\|X_u - X_v\|_{\psi_2} \leq \frac{\sigma}{k}\left(\sum_{i=1}^k \langle \Phi(x_i), u-v \rangle^2\right)^{1/2} \leq \frac{\sigma C_{\Phi,\infty}}{\sqrt{k}}\|u-v\|_1 \leq \frac{\sigma C_{\Phi,\infty}\sqrt{m}}{\sqrt{k}}\|u-v\|_2.$$

Hence, by Dudley's entropy integral (applied with the $L_2$ metric on $\mathcal{U}_m$ and $\mathrm{diam}(\mathcal{U}_m) \leq 2$),

$$\mathbb{E}\left[\|Z_\varepsilon\|_{2,m}^*\right] \lesssim \frac{\sigma C_{\Phi,\infty}}{\sqrt{k}}\int_0^{\mathrm{diam}(\mathcal{U}_m)} \sqrt{\log \mathcal{N}(\mathcal{U}_m, \|\cdot\|_2, \varepsilon)}\, d\varepsilon.$$

Using the standard bound $\mathcal{N}(\mathcal{U}_m, \|\cdot\|_2, \varepsilon) \leq \binom{d_1}{m}(3/\varepsilon)^m$,

$$\mathbb{E}\left[\|Z_\varepsilon\|_{2,m}^*\right] \lesssim \frac{\sigma C_{\Phi,\infty}}{\sqrt{k}}\int_0^2 \sqrt{m\log(d_1/m) + m\log(3/\varepsilon)}\, d\varepsilon \lesssim \sigma C_{\Phi,\infty}\sqrt{\frac{m\log(d_1/m)}{k}}.$$

We conclude using a standard concentration inequality for the supremum $W = \|Z_\varepsilon\|_{2,m}^*$ of a subgaussian process: with probability at least $1 - \delta$,

$$\|Z_\varepsilon\|_{2,m}^* \lesssim \mathbb{E}[\|Z_\varepsilon\|_{2,m}^*] + \sigma C_{\Phi,\infty}\sqrt{\frac{m\log(1/\delta)}{k}} \lesssim \sigma C_{\Phi,\infty}\left(\sqrt{\frac{m\log(d_1/m)}{k}} + \sqrt{\frac{m\log(1/\delta)}{k}}\right).$$

The result follows by substituting $m = 2s'$ and using $\sqrt{a} + \sqrt{b} \leq \sqrt{2(a+b)}$ to consolidate terms under a single universal constant $C_H$. $\square$

**Lemma 7** (Misspecification correlation bound). *Let $\Delta$ be a misspecification vector such that $\frac{1}{k}\|\Delta\|_2^2 \leq \eta$. Under A7 ($L_\infty$ bound), the correlation of $\Delta$ with sparse concept vectors is bounded by:*

$$\left\|\frac{1}{k}\boldsymbol{\Phi}_{x^\star}^\top\Delta\right\|_{2,2s'}^* \leq \eta_\Delta := \sqrt{2s'}\,C_{\Phi,\infty}\sqrt{\eta}.$$

*Proof.* Let $Z_\Delta = \frac{1}{k}\boldsymbol{\Phi}_{x^\star}^\top\Delta$. We first bound the $L_\infty$ norm.

$$\|Z_\Delta\|_\infty = \max_j \left|\frac{1}{k}\sum_{i=1}^k \Phi_{ij}\Delta_i\right|.$$

By Cauchy-Schwarz, $|\sum_i \Phi_{ij}\Delta_i| \leq \sqrt{\sum_i \Phi_{ij}^2}\sqrt{\sum_i \Delta_i^2} \leq \sqrt{kC_{\Phi,\infty}^2}\sqrt{k\eta}$. Thus, $\|Z_\Delta\|_\infty \leq C_{\Phi,\infty}\sqrt{\eta}$. The sparse dual norm is bounded by:

$$\|Z_\Delta\|_{2,2s'}^* = \max_{S:|S|=2s'} \|(Z_\Delta)_S\|_2 \leq \sqrt{2s'}\|Z_\Delta\|_\infty \leq \sqrt{2s'}C_{\Phi,\infty}\sqrt{\eta}.$$

$\square$

### C.3 INSUFFICIENCY OF GLOBAL TRAINING

We define a specific construction that satisfies A1–A5 and models the superposition of concepts through a randomized feature map. This construction captures the challenge faced by an underparameterized model learning a complex environment.

**Definition 8** (Globally non-learnable instance). *Let $d_2 \geq \Omega(s \log d_1)$. We assume the following:*

1.  Data distribution and concepts: *We partition the input space $\mathcal{X}$ into $M = d_1$ disjoint neighborhoods $\{B_m\}_{m=1}^{d_1}$ with equal probability $\mathbb{P}(B_m) = 1/d_1$. The concept map is constant locally: $\Phi(x) = e_m \in \mathbb{R}^{d_1}$ (standard basis vector) for $x \in B_m$. This is 1-sparse.*

2.  Ground truth: *The observations are noiseless and constant $y = 1$ everywhere. The global ground truth vector is $w_\star = \mathbf{1}$. Locally, the function is perfectly matched by the 1-sparse model $w_m = e_m$.*

3.  Learned features & random superposition: *The feature map $\Psi : \mathcal{X} \to \mathbb{R}^{d_2}$ is defined such that $\Psi(x) = p_m$ for $x \in B_m$, which implies local linearity of features. We model the learned features by assuming the representations $\{p_m\}_{m=1}^{d_1}$ are drawn independently and uniformly from the unit sphere $\mathcal{S}^{d_2-1}$.*

We first verify that Definition 8 satisfies Assumptions 1 to 5, A7, A8, leading to a misspecification error of $\eta_{\text{mis}} = 0$. As a consequence and by Proposition 4, TTT is consistent. Notably, the dimension of the feature map $d_2$ may be exponentially smaller than the number of concepts $d_1$, yet TTT can still learn the ground truth perfectly.

*Proof.*

A1 (Sparse concepts): $\Phi(x) = e_m$ is 1-sparse.

A7 (Bounded concepts): $\|\Phi(x)\|_2 = 1$.

A8 (Linear model): $y = 1$. $w_* = \mathbf{1}$. $\sigma^2 = 0$.

A2 (Neighborhood preservation): We require $|\text{sim}_\Psi(x, x') - \text{sim}_\Phi(x, x')| \leq \eta_{\text{ang}}$. Consider $x \in B_i, x' \in B_j$ with $i \neq j$. $\text{sim}_\Phi(x, x') = \langle e_i, e_j \rangle = 0$. $\text{sim}_\Psi(x, x') = \langle p_i, p_j \rangle$. Since $p_i, p_j$ are drawn uniformly from $\mathcal{S}^{d_2-1}$, the inner product concentrates around 0. By standard concentration inequalities (related to the Johnson-Lindenstrauss lemma), with high probability over the draw of all pairs $\{p_i\}$, we have $\max_{i \neq j} |\langle p_i, p_j \rangle| \leq O(\sqrt{\log(d_1)/d_2})$. Thus, A2 holds with $\eta_{\text{ang}}$ small if $d_2$ is sufficiently large compared to $\log(d_1)$.

A3 (Local simplicity): In $B_m$, $f(x) = 1$. $w_m = e_m$ is 1-sparse and achieves $\eta_{\text{spa}} = 0$ since $\langle \Phi(x), w_* \rangle = \langle e_m, \mathbf{1} \rangle = 1$ and $\langle \Phi(x), w_m \rangle = \langle e_m, e_m \rangle = 1$.

To verify A4, A5, we need to define the local linear maps $P_m$.

A4 (Local linearity): We require $\Psi(x) = P_m \Phi(x)$ for $x \in B_m$. This means $p_m = P_m e_m$. We construct $P_m \in \mathbb{R}^{d_2 \times d_1}$ by setting the $m$-th column to $p_m$ and all other columns to zero: $P_m = [0, \ldots, p_m, \ldots, 0]$.

A5 (Expressivity): We require $w_m = e_m$ to be in the row space of $P_m$. The row space of the constructed $P_m$ is exactly $\text{span}(e_m)$. Thus, $\eta_{\text{rep}} = 0$. We also need to verify that the optimal local solution corresponds to a sparse concept vector. The local optimization is $\min_v (1 - \langle p_m, v \rangle)^2$. The minimum norm solution is $v_m^* = p_m / \|p_m\|^2 = p_m$ (since $\|p_m\| = 1$). We check the sparsity of the corresponding concept vector $P_m^\top v_m^* = P_m^\top p_m$. The $j$-th component of $P_m^\top p_m$ is $\langle \text{col}_j(P_m), p_m \rangle$. For $j = m$, it is $\langle p_m, p_m \rangle = 1$. For $j \neq m$, it is $\langle 0, p_m \rangle = 0$. Thus, $P_m^\top v_m^* = e_m$, which is 1-sparse.

$\square$

We compare this to training a single global model on all data,

$$\hat{v}^{\text{global}} := \underset{v \in \mathbb{R}^{d_2}, \|v\|_2 \text{ minimized}}{\arg\min} \frac{1}{N} \sum_{i=1}^{N} (y_i - \langle \Psi(x_i), v \rangle)^2.$$

Remarkably, global training fails to learn this "simple" ground truth function, even as $N \to \infty$. Due to being underparameterized, the model's features represent concepts in superposition, i.e., the features $p_m$ are not orthogonal. Thus, adjusting the global model to fit one neighborhood inevitably interferes with the predictions in other neighborhoods. Global training therefore has to find a compromise that minimizes the average error across all neighborhoods.

**Proposition 9** (Interference error of global training)**.** *Consider the instance of Definition 8. The expected approximation error of the global model, averaged over the random realizations of the feature map $\Psi$, is $\mathbb{E}_\Psi[(y - \langle \Psi(x), \hat{v}^{\text{global}} \rangle)^2] = 1 - \frac{d_2}{d_1}$.*

*Remark* 10. Note that if the global model is not underparameterized, i.e., $d_2 = d_1$, the error of global training is zero, as one would naturally expect. On the other hand, the trivial global model $\hat{v}^{\text{global}} = \mathbf{0}$ has error 1. As the model size $d_2$ shrinks, the error of global training increases towards 1. As the number of distinct concepts $d_1$ increases, the error of global training also increases, approaching 1 as $d_1 \to \infty$. When increasing the number of distinct concepts, global training must compromise between more neighborhoods, leading to higher interference.

*Proof of Proposition 9.* We analyze the approximation error of the global model. The global loss is:

$$L^{\text{global}}(v) = \mathbb{E}[(y - \langle \Psi(x), v \rangle)^2] = \frac{1}{d_1} \sum_{m=1}^{d_1} (1 - \langle p_m, v \rangle)^2. \tag{11}$$

Let $P \in \mathbb{R}^{d_1 \times d_2}$ be the matrix whose rows are $p_m^\top$. The loss can be written in vector form:

$$L^{\text{global}}(v) = \frac{1}{d_1} \|\mathbf{1} - Pv\|^2. \tag{12}$$

This is a standard least squares problem. We assume $d_1 > d_2$. Since the vectors $p_m$ are drawn from a continuous distribution (uniform on the sphere), $P$ has full column rank ($d_2$) with probability 1.

The optimal global model is $\hat{v}^{\text{global}} = (P^\top P)^{-1} P^\top \mathbf{1}$. The resulting approximation error is:

$$\mathcal{E}^{\text{global}} := \mathbb{E}[(y - \langle \Psi(x), \hat{v}^{\text{global}} \rangle)^2] = L^{\text{global}}(\hat{v}^{\text{global}}) = \frac{1}{d_1} \|\mathbf{1} - P(P^\top P)^{-1} P^\top \mathbf{1}\|^2.$$

Let $\Pi = P(P^\top P)^{-1} P^\top \in \mathbb{R}^{d_1 \times d_1}$ be the orthogonal projection matrix onto the column space of $P$. The residual vector is $\mathbf{1} - \Pi \mathbf{1} = (I - \Pi)\mathbf{1}$. Since $(I - \Pi)$ is also an orthogonal projection matrix, $(I - \Pi)^\top (I - \Pi) = (I - \Pi)$.

$$\begin{aligned} \mathcal{E}^{\text{global}} &= \frac{1}{d_1} \mathbf{1}^\top (I - \Pi)^\top (I - \Pi) \mathbf{1} = \frac{1}{d_1} \mathbf{1}^\top (I - \Pi)\mathbf{1} \\ &= \frac{1}{d_1} (\mathbf{1}^\top \mathbf{1} - \mathbf{1}^\top \Pi \mathbf{1}) = \frac{1}{d_1}(d_1 - \mathbf{1}^\top \Pi \mathbf{1}) \\ &= 1 - \frac{1}{d_1} \mathbf{1}^\top \Pi \mathbf{1}. \end{aligned}$$

We want to calculate the expectation of this error over the random realization of $P$.

$$\mathbb{E}[\mathcal{E}^{\text{global}}] = 1 - \frac{1}{d_1} \mathbb{E}[\mathbf{1}^\top \Pi \mathbf{1}]. \tag{13}$$

We next analyze the term $\mathbf{1}^\top \Pi \mathbf{1} = \sum_{i=1}^{d_1} \sum_{j=1}^{d_1} \Pi_{ij}$ and the expected values of the entries $\mathbb{E}[\Pi_{ij}]$.

**Step 1: Diagonal elements ($i = j$).** $\Pi_{ii}$ is the leverage score of the $i$-th data point $p_i$. The trace of a projection matrix equals its rank. Since $P$ has rank $d_2$ (w.p. 1), $\text{Tr}(\Pi) = d_2$.

$$\text{Tr}(\Pi) = \sum_{i=1}^{d_1} \Pi_{ii}. \tag{14}$$

Taking the expectation:

$$\mathbb{E}[\mathrm{Tr}(\Pi)] = \sum_{i=1}^{d_1} \mathbb{E}[\Pi_{ii}] = d_2. \tag{15}$$

Since the vectors $\{p_m\}$ are drawn i.i.d., the distribution of $P$ is invariant under permutation of the rows. Thus, $\mathbb{E}[\Pi_{ii}]$ must be the same for all $i$.

$$d_1 \mathbb{E}[\Pi_{ii}] = d_2 \implies \mathbb{E}[\Pi_{ii}] = \frac{d_2}{d_1}. \tag{16}$$

**Step 2: Off-diagonal elements ($i \neq j$).** We show that $\mathbb{E}[\Pi_{ij}] = 0$ using a symmetry argument. The entry $\Pi_{ij}$ is given by $p_i^\top (P^\top P)^{-1} p_j$. Let $S = P^\top P$.

We examine the conditional expectation $\mathbb{E}_{p_i}[\Pi_{ij}|\{p_k\}_{k \neq i}]$. Let $S_{-i} = \sum_{k \neq i} p_k p_k^\top$ and $S = S_{-i} + p_i p_i^\top$. Since $d_1 > d_2$, we have $d_1 - 1 \geq d_2$. As the distribution is continuous, $S_{-i}$ is invertible (rank $d_2$) with probability 1.

We use the Sherman-Morrison formula to analyze $\Pi_{ij} = p_i^\top S^{-1} p_j$.

$$S^{-1} = S_{-i}^{-1} - \frac{S_{-i}^{-1} p_i p_i^\top S_{-i}^{-1}}{1 + p_i^\top S_{-i}^{-1} p_i}. \tag{17}$$

Applying $p_i^\top$ from the left and $p_j$ from the right:

$$\Pi_{ij} = p_i^\top S_{-i}^{-1} p_j - \frac{(p_i^\top S_{-i}^{-1} p_i)(p_i^\top S_{-i}^{-1} p_j)}{1 + p_i^\top S_{-i}^{-1} p_i}$$

$$= (p_i^\top S_{-i}^{-1} p_j) \left( 1 - \frac{p_i^\top S_{-i}^{-1} p_i}{1 + p_i^\top S_{-i}^{-1} p_i} \right)$$

$$= \frac{p_i^\top S_{-i}^{-1} p_j}{1 + p_i^\top S_{-i}^{-1} p_i}.$$

Let $A = S_{-i}^{-1}$ (which is positive definite) and $u = S_{-i}^{-1} p_j$. Note that $A$ and $u$ are independent of $p_i$. We define the function $h(p_i) = \frac{\langle p_i, u \rangle}{1 + \langle p_i, A p_i \rangle}$.

We observe that $h(p_i)$ is an odd function of $p_i$:

$$h(-p_i) = \frac{\langle -p_i, u \rangle}{1 + \langle -p_i, A(-p_i) \rangle} = \frac{-\langle p_i, u \rangle}{1 + \langle p_i, A p_i \rangle} = -h(p_i). \tag{18}$$

The distribution of $p_i$ (uniform on the sphere) is symmetric around the origin. The expectation of an odd integrable function over a symmetric distribution is zero.

$$\mathbb{E}_{p_i}[\Pi_{ij}|\{p_k\}_{k \neq i}] = \mathbb{E}_{p_i}[h(p_i)] = 0. \tag{19}$$

By the law of total expectation, $\mathbb{E}[\Pi_{ij}] = 0$ for $i \neq j$.

**Step 3: Finalizing.** We combine the results for the diagonal and off-diagonal elements:

$$\mathbb{E}[\mathbf{1}^\top \Pi \mathbf{1}] = \sum_i \mathbb{E}[\Pi_{ii}] + \sum_{i \neq j} \mathbb{E}[\Pi_{ij}]$$

$$= d_1 \cdot \frac{d_2}{d_1} + d_1(d_1 - 1) \cdot 0 = d_2.$$

Finally, the expected global error is:

$$\mathbb{E}[\mathcal{E}^{\mathrm{global}}] = 1 - \frac{1}{d_1} \mathbb{E}[\mathbf{1}^\top \Pi \mathbf{1}] = 1 - \frac{d_2}{d_1}. \tag{20}$$

$\square$

C.4    HARDNESS OF GLOBAL LEARNING UNDER THE LRH: MAIN PROPOSITION AND PROOF

We begin by stating the formal version of Proposition 2:

**Proposition 11.** *Let $c_1 \geq 0$, $\sigma \lesssim 1$, $k \asymp d^{c_1}$, and $s \asymp 1$. Assume $n \gtrsim \exp(\Omega(\log d)^2)$. Then Equation* (4) *holds and there exists a polynomial-time algorithm $\mathcal{A}$ in its input, such that*

$$\sup_{f^\star \in \mathcal{F}_{d,s,k}} \mathbb{E}_{\mathcal{D}} \|\mathcal{A}(\mathcal{D}) - f^\star\|_{L_2(\gamma_d)} \leq 0.1.$$

*Under the low-degree polynomial conjecture, this bound is tight, $n \gtrsim \exp(\Omega(\log d)^2)$ samples are required by any algorithm that relies on the Statistical Query (SQ) and the Low-Degree Polynomial (LDP) frameworks. Furthermore, without the low-degree polynomial conjecture, one can show that* $\text{poly}(n)$ *samples are needed for any algorithm.*

For further details on the Statistical Query (SQ) and the Low-Degree Polynomial (LDP) frameworks, we refer to Reyzin (2020).

*Proof.* Recalling the definition of $\mathcal{F}_{d,s,k}$ and using that $s \asymp 1$ and $\|f^\star\|_{L_2(\gamma_d)} \asymp 1$, we obtain that

$$\sum_{i=1}^{d_1} \|\partial_i f^\star\|_1^2 \lesssim \frac{s}{d_1} \lesssim \log(1/d_1)\|f^\star\|_{L_2(\gamma_d)}^2.$$

Paouris et al. (2022) showed

$$\text{Var}(P_t[f^\star]) \leq \exp(2-2t) \cdot \text{Var}(f^\star) \cdot \exp(-ct\log(d)) \lesssim \exp(-ct\log(d)), \tag{21}$$

where $P_t$ is the Ornstein-Uhlenbeck (OU) semigroup, i.e.

$$P_t[f^\star](x) = \mathbb{E}_{Z \sim \gamma_n}[f^\star(\exp(-t)x + \sqrt{1-\exp(-2t)}Z] = \sum_{m=0}^\infty \exp(-tm)\mathcal{P}_m(f^\star),$$

and here $\mathcal{P}_m(f^\star)$ is the projection operator on the $m$-Hermite polynomials.

Using Equation (21), we conclude that for $m \in 1, \ldots, c\log(d)$ (by choosing $t_m \asymp 1/m$)

$$\|\mathcal{P}_m(f^\star)\|_{L_2(\gamma_d)}^2 \lesssim \exp(-c_1\log(d)/m)).$$

Now, recall that we assume $k$ is polynomial in $d$, and note that up to a measure zero, for $\Delta_t := P_t[f^\star] - f^\star$, it holds for any $t \ll 1$ that

$$\forall x \in \mathbb{R}^d \quad \Delta_t(x) \lesssim 2\sqrt{\log(k)} \cdot s\sqrt{t} \lesssim \sqrt{\log(d_1) \cdot t},$$

where we used our assumption that there are $k$ cells, the definition of the OU group, and the maximal inequality of $2k$-Gaussian. Therefore, we can choose $t \leq c/\log(d)$, for small enough $c \geq 0$ and obtain that

$$\|P_{c/\log(d/n)}f^\star\|_{L_2(\gamma_d)} \geq 0.99.$$

If there were more than $0.1$ $L_2^2(\gamma_n)$, energy in the $m$-coefficients for $m \geq C\log(d/n)$ and $C \geq 0$ large enough, we would obtain a contraction. As for $t = c\log(d)$, it holds that

$$\exp(-mt) \cdot \|\mathcal{P}_m(f^\star)\|_{L_2(\gamma_d)}^2 \lesssim \exp(-cm/\log(d))\|f^\star\|_{L_2(\gamma_d)}^2 \leq 0.01 \cdot \|\mathcal{P}_m(f^\star)\|_{L_2(\gamma_d)}^2,$$

which cannot align with the previous equation and our assumption of $\|f^\star\|_{L_2(\gamma_n)} = 1$. In words, this property says that $f^\star$ is far from being a "low degree" polynomial. Meaning that

$$0.1 \leq \left\|\mathbb{E}f^\star + \sum_{c\log(d/s) \leq m \leq C\log(d)} \mathcal{P}_m(f^\star)\right\|_{L_2(\gamma_d)}^2 \leq 0.9,$$

where $c, c_1 \in (0,1)$, $C \geq 1$ are absolute constants.

Therefore, to obtain a $0.1$ approximation to $f^\star$, we need to learn the coefficients of at most (and at least) the top $\Theta(\log(d))$ basis of the Hermite polynomials. Using the result of Bizeul & Klartag (2025) (or the classical work of Kalai et al. (2008)), it can be done with

$$n \lesssim s^2 \log(k) \cdot d^{C\log(k)} \asymp s^2 \log(k) \cdot k^{\log(d)s^2} \asymp \exp(C\log(d)^2)$$

samples. By definition, one can easily see that $\mathcal{F}_{d,s,k}$ is much harder to learn than the Gaussian Index Model (GIM). Since the generative component of our functions satisfies $k^* \asymp \log(d/n)$, it follows that, under the low-degree polynomial conjecture, these classes require at least $\exp(\Theta(\log(d/n)^2))$ samples to even learn the GIM, see Damian et al. (2024) and references within, and the seminal work of Arous et al. (2021).

Without the low-degree polynomial conjecture, one may use the result of (Klivans et al., 2008, Thms. 26 and 27), which shows that the learnability of the subclass of

$$\{1_K(x) : K \text{ is polytope in } \mathbb{R}^d \text{ with } \mathrm{Poly}(d) \text{ facets}\} \subset \{\mathbb{R}^d \to \mathbb{R}\}$$

requires at least $\exp(\Omega(\log(d)) \asymp \mathrm{Poly}(d)$ samples in the information theoretic sense. However, an algorithmic gap remains in these classical works Klivans et al. (2008); Kalai et al. (2008), and there is also a gap in the corresponding upper bound for the sample complexity.

$\square$

## D    ADDITIONAL EXPERIMENTAL RESULTS

This section provides additional experimental results that complement the findings presented in the main body.

### D.1    SCALING WITH FEATURE DIMENSION

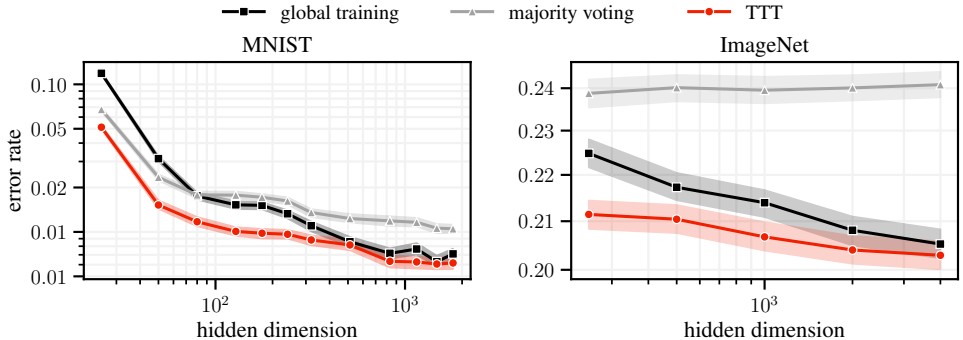

Figure 6: **Scaling model width.** Classification error rates when scaling model width. We evaluate a globally trained model (black) across different model sizes, as well as TTT (red) and a majority vote on the neighborhood (gray). While majority vote leads to a poor predictor with many classes (i.e., complex tasks), TTT consistently outperforms global training, with the performance gap shrinking as the model size increases.

For the scaling experiments on MNIST and ImageNet shown in Figure 4, we varied only the network width while keeping the depth fixed, and we reported the resulting error rates as a function of the total number of model parameters. Our theoretical analysis (§5) relies on the expressivity of the feature space, which requires the feature dimension $d_2$ to scale at least proportionally to $s \log d_1$. To make this connection explicit, we present the same results from the main body as a function of the feature dimension in Figure 6 (instead of the total parameter count). The resulting curves show similar qualitative and quantitative behavior: TTT consistently outperforms global training and majority vote, with the performance gap narrowing as the hidden dimension increases.

### D.2    END-TO-END TEST-TIME TRAINING

Because full end-to-end training on the neighborhood set is computationally expensive, it is common to fine-tune only the last linear layer (or the last few layers) as a trade-off between computational cost and performance. An alternative is to apply a parameter-efficient fine-tuning method such as LoRA (Hu et al., 2022). In the experiments in Figure 4 for the image tasks (MNIST, ImageNet), we adopted the former approach and fine-tuned only the classification head. For the language modeling experiments, we used LoRA to fine-tune the model. Both strategies introduce a capacity constraint

on the TTT model. To test whether our conclusions still hold when removing this constraint, we conduct an end-to-end TTT experiment on MNIST.

Mirroring the scaling experiment in Figure 4, we train LeNet (LeCun et al., 1998a) convolutional neural networks (CNNs) of varying sizes across five random seeds. Details about the global pre-training procedure are provided in Appendix F.4. Based on these pre-trained models, we compare two variants of TTT. First, we apply last-layer TTT, where all layers except the final one are frozen and only the classification head is fine-tuned on the neighborhood set. This matches the setup used in Figure 4. Second, we perform end-to-end TTT: neighbors are selected using the $L_2$-distance in the last-hidden-layer representations of the pre-trained model (as before), but subsequently all model weights are fine-tuned. The resulting model is then used to predict the class label of the test sample. The fine-tuning hyperparameters for end-to-end TTT are provided in Table 3, obtained by tuning on the validation set.

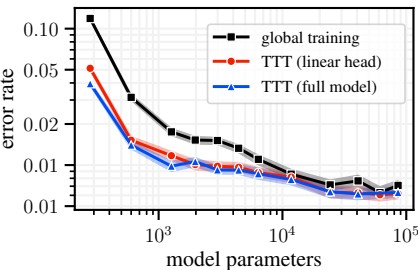

Figure 7: Classification error rates when scaling model width on MNIST. We compare last-layer TTT (red) against end-to-end TTT (blue) with the globally trained model (black) serving as the baseline. Both TTT variants yield nearly identical error rates.

| | Hidden dimension | | | | | | | | | | |
|---|---|---|---|---|---|---|---|---|---|---|---|
| | **25** | **50** | **80** | **128** | **176** | **240** | **320** | **512** | **832** | **1152** | **1792** |
| Learning rate | 5e-3 | 0.01 | 0.01 | 0.01 | 5e-3 | 1e-3 | 1e-3 | 1e-3 | 1e-3 | 5e-4 | 5e-4 |
| Epochs | 200 | 200 | 300 | 100 | 200 | 200 | 200 | 300 | 300 | 300 | 200 |
| Neighbors | 300 | 50 | 200 | 300 | 300 | 50 | 50 | 300 | 300 | 50 | 300 |

Table 3: Hyperparameters for end-to-end TTT.

# E    ADDITIONAL ABLATIONS

While the improvement of TTT in image classification may seem small in terms of classification error, we find that TTT can significantly reduce cross-entropy loss on the test set, as shown in Figure 8.

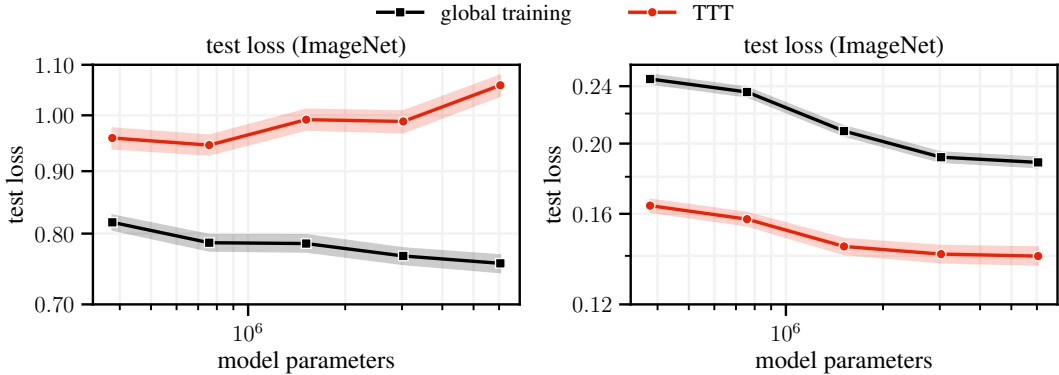

Figure 8: **Left:** The average cross-entropy loss of the test samples of the globally trained MLP head is lower compared to the test-time training (TTT) model. **Right:** TTT substantially lowers cross-entropy loss on the test point compared to the global model when, to filter noisy labels, we filter to test points where either the global model or TTT predict the correct label. This suggests that TTT increases cross-entropy significantly on the noisy labels that are not predictable by the base model, while lowering cross-entropy on the points which are predictable.

# F EXPERIMENT DETAILS

In this section, we provide additional details about our experimental setup. These include the experiments used to validate sparsity (§3) as well as those designed to analyze and illustrate the implications of our theoretical results (§4).

We open-source our code implementation.[9]

## F.1 COMPARISON OF LOGITS (FIGURE 3)

We start by introducing the related notation. Define the top-10 probabilities for TTT in the sparse concept space for the test point $x^\star$ as a vector $p^{\hat\Phi}(x^\star) \in [0,1]^{10}$, similarly, $p^{\hat\Psi}(x^\star)$ for dense reconstructions. In this notation, we implicitly assume that the probabilities $p^{\hat\Phi}(x^\star)$ are scaled by some optimal factor $\tau_{x^\star}$. In particular, define the logits corresponding to $p^{\hat\Phi}(x^\star)$ (i.e., before the softmax application) by $\mathrm{logit}^{\hat\Phi}(x^\star)$. Then, for each test point $x^\star$, we "calibrate" (cf. (Xie et al., 2024)) the concept space probabilities $p^{\hat\Phi}(x^\star)$ by finding the optimal scale $\tau_{x^\star}$ which aligns them closer. Namely, we choose $\tau_{x^\star}$ to minimize the following KL-divergence term:

$$\mathrm{KL}\left(p^{\hat\Psi}(x^\star) \,\Big\|\, \bar{p}^{\hat\Phi}(x^\star)\right) \to \min_{\tau_{x^\star}},$$

$$\bar{p}^{\hat\Phi}(x^\star) := \mathrm{softmax}\left(\frac{\mathrm{logit}^{\hat\Phi}(x^\star)}{\tau_{x^\star}}\right).$$

We note that although the temperature $\tau_{x^\star}$ is adjusted on a sample-by-sample basis, the variation is not significant. Specifically, we find the temperature has a mean of around $0.8$ and a standard deviation of around $0.1$. This mean value ($\tau_{x^\star} < 1$) indicates that the predictive distribution $p^{\hat\Psi}(x^\star)$ in the dense space has a slightly sharper profile than the baseline.

We now introduce a metric to quantify the remaining discrepancy between the calibrated sparse distribution and the dense distribution. The relative total variation at position $i \in \{1, \ldots, 10\}$ is defined as follows:

$$\mathrm{relTV}_i(\hat\Psi, \hat\Phi) := \mathbb{E}_{x^\star}\left[\frac{\left|p_i^{\hat\Psi}(x^\star) - p_i^{\hat\Phi}(x^\star)\right|}{\frac{1}{2} \cdot \left(\mathbb{E}_{x^\star}[p_i^{\hat\Psi}(x^\star)] + \mathbb{E}_{x^\star}[p_i^{\hat\Phi}(x^\star)]\right)}\right],$$

where the absolute difference is normalized by the average magnitude of the corresponding probabilities. Incorporating both averages into the denominator ensures the robustness of the quantity to noisy observations, since both the scale of $p_i^{\hat\Phi}$ and $p_i^{\hat\Psi}$ are taken into account.

## F.2 SAE ON MNIST DATA

We evaluate the SAE setup on the MNIST dataset (10k test samples, 60k training samples). To obtain a sparse concept vector $\hat\Phi(x_i)$ for each image $x_i \in \mathcal{D}_{\mathrm{MNIST}}$, we employ a Gemma-Scope-style SAE (Lieberum et al., 2024). The only conceptual difference from regular SAE applications is that our encoder has a convolutional architecture, i.e., it is LeNet-like (LeCun et al., 1998a) and maps each image to a representation vector $\Psi(x_i) \in \mathbb{R}^{d_2}$ with $d_2 = 256$, which is then lifted to a sparse concept vector $\hat\Phi(x_i)$. However, the decoder is still *linear* to force linearity of the concept space. We also directly reconstruct the inputs, i.e., images $x$, which means that reconstruction error takes the following form:

$$\mathbb{E}_{x \in \mathcal{D}}\|x - D \cdot \hat\Phi(x)\|_2, \quad D \in \mathbb{R}^{28 \cdot 28 \times d_1}.$$

In this setup, $\hat\Phi(x_i) \in \mathbb{R}^{d_1}$ where $d_1 = 1024$ and the average sparsity of the concept vector is

$$\mathbb{E}_x[\|\hat\Phi(x)\|_0] \approx 18.9.$$

---

[9] https://github.com/patrikwolf/ttt_theory

In the Gemma-Scope-style SAE, the sparsity constraint is enforced via a thresholding activation, i.e., inputs $v \in \mathbb{R}^{d_1}$ are passed through

$$\tilde{v} := \mathrm{ReLU}(v - \theta),$$

where the ReLU is applied component-wise for some thresholds $\theta \in \mathbb{R}^{d_1}$ independent of $v$. Consequently, all active components of the sparse vector $\tilde{v}$ are positive. In addition to thresholding, the Gemma-Scope-style SAE implicitly enforces the desired sparsity level via an $L_0$ penalty on concept vectors $\hat{\Phi}(x)$ and uses straight-through estimator for the gradient estimate.

The procedure of searching for the optimal mask $m$ stays the same as per Section 3, however, we note that in this setup the penalty $\lambda$ is set to $10^{-3}$. For this experiment, the average resulting sparsity of the mask $m$ is $\mathbb{E}_{x^\star}[\|m\|_0] \approx 24.5$.

For a dense base model (i.e., a counterpart of CLIP embeddings in the case of ImageNet), we train a variant of LeNet (LeCun et al., 1998a) with a scale of 0.5. The resulting feature dimension before the final linear classification layer is equal to 50. This model is clearly underparameterized with test accuracy 94.51 %.

For the base CNN TTT model, we perform 100 full-batch steps of Adam with learning rate of $10^{-1}$ (increasing budget does not affect the performance). For TTT with the adaptive mask in the concept space, we do 200 full-batch steps of Adam with learning rate $5 \cdot 10^{-2}$. The neighborhood size for TTT is set to $n = 100$.

The resulting accuracies are:

- LeNet CNN TTT: 97.62,
- Masked SAE TTT: 97.72.

**Auxiliary observations.** We use the same notation for active sets for concept vectors as in Section 3. However, for better clarity, we recap the notation. For each sparse vector $\Phi(x) \in \mathbb{R}^{d_1}$ we define its active set as the set of vector components that are non-zero, i.e.,

$$m(\Phi(x)) = \{\ell : (\Phi(x))_\ell > 0\}.$$

In this spirit, we define the active set for the current test point $x^\star$ as follows:

$$m_{x^\star} := m(\Phi(x^\star)).$$

Similarly, active components of neighbors are defined as

$$m_i := m(\Phi(x_i)), \quad x_i \in \mathcal{D}_{x^\star}^\Phi.$$

In this context, the "intersection" analysis reveals the following properties of the adaptive mask $m$, active set of the test point $m_{x^\star}$ and neighbors' active sets $m_i$:

- $\mathbb{E}_x | \cup_i m_i | \approx 144.5, \quad \mathbb{E}_x | m_{x^\star} \cap m | \approx 7.9,$
- $\mathbb{E}_x | \cup_i (m_i \cap m) | \approx 7.1, \quad \mathbb{E}_x | \cup_i (m_i \cap m_{x^\star}) | \approx 8.3.$

In particular, one might be tempted to draw the following simplifying conclusion:

$$m \equiv \cup_i (m_i \cap m_{x^\star}),$$

that is, to define the mask $m$ so that it only selects indices appearing in both the test point and its neighbors. However, this approach fails on more complex datasets (e.g., ImageNet, see Section 3), because additional slack components in the mask are necessary to capture "non-spurious" test features.

### F.3   SAE ON IMAGENET CLIP EMBEDDINGS

**Top-$k$ SAE training.** Obtaining a sparse autoencoder with meaningful features (with mild amount of non-active neurons) is a task with multiple caveats. We employ several common techniques to improve the training procedure, which we describe below.

We use a learning rate warm-up to ensure that the concept space is properly explored at the start of training, preventing neurons from deactivating and becoming trapped in suboptimal configurations.

We warm up the learning rate *linearly* to the value $3 \cdot 10^{-4}$ for $T_0 = 5000$ steps. After this, we employ a typical cosine decay with a horizon of $T = 10^5$. In particular, let $i$ be the current step, then at each step the initial learning (in this case $3 \cdot 10^{-4}$) is multiplied by the value of $\lambda_i$, which is computed as follows:

$$\lambda_i := 0.5 \cdot \left( 1 + \cos \left( \pi \cdot \tilde{\lambda} \right) \right) \qquad \text{with} \qquad \tilde{\lambda}_i := (i - T_0)/(T - T_0).$$

In addition to the learning rate schedule, we also gently ramp up the sparsity of the concept vector to the desired value of $k = 16$ as follows: let $k_0 = 128$ be the initial sparsity of the concept vector and $K = 10000$ be the number of warm-up steps, then

$$k_i := k_0 - (k_0 - k) \cdot \gamma_i \qquad \text{with} \qquad \gamma_i := i/K.$$

Once the target sparsity of $k = 16$ is reached, the value remains at the respective level until the end of the training.

We now describe the implementation of the ghost gradients (Gao et al., 2025) used in our ImageNet run. In a nutshell, ghost loss ensures that features that are not activated in the top-$k$ are still getting learning signal during the training. This is very important as non-convexity of the optimization landscape often leads to a suboptimal configuration for which considerable amount of units is inactive. The ghost gradient method aims at "shaking" these units up to make them active again. Thus, we first need to define which units are considered inactive during the training. Let $f_i \in \mathbb{R}^{d_1}$ denote the vector of unit activation frequencies after iteration $i$, where

$$(f_i)_j = \frac{\text{number of times unit } j \text{ is active after } i \text{ processed samples}}{i}.$$

Then the unit $j$ is considered "almost inactive" if $(f_i)_j \leq 10^{-4}$. Thus, we define the corresponding "inactivity" binary mask as $(m_i)_j = \mathbb{I}\{(f_i)_j \leq 10^{-4}\}$. Ghost gradient uses these features to improve the current reconstruction $\hat{\Psi}(x) - \Psi(x)$. Namely, the inactive features are decoded, i.e., $\widetilde{\Psi}(x) = D \cdot (m_i \odot (E \cdot \Psi(x)))$, as if they were present and used to minimize:

$$\frac{1}{d_2} \cdot \|\hat{\Psi}(x) - \Psi(x) - \widetilde{\Psi}(x)\|_2^2. \tag{22}$$

We add the ghost loss (22) to the initial reconstruction objective with weight of $10^6$.

We also employ gradient clipping for more stable iterations to the norm range of $[0, 1]$, and introduce additional dropout with rate $0.5$ for the pre-activations $E \cdot \Psi(x)$ to foster the diversity of concepts. We use column-wise normalization for the decoder weights to enjoy more stable training. Note that, since CLIP embeddings have unit norm, such restriction does not hinder the expressivity of the decoder. We also initialize the decoder to be the transpose of the encoder, which is common practice in SAEs. As for generic hyperparameters for the Adam optimizer (Kingma & Ba, 2015), we fix them to the following values: batch size of $4096$, weight decay of $0$, number of epochs $100$, and Adam's $\beta_1$ of $0.9$ and $\beta_2$ of $0.999$.

**Global training and TTT.** To train the global concept space and CLIP reconstruction models, we perform 100 epochs of batch size 512 using Adam optimizer with learning rate 0.001 and weight decay of $5 \cdot 10^{-9}$. For each TTT point, we do 80 full-batch steps (batch size 50) of Adam with learning rate 0.02 and zero weight decay. The latter hyperparameter set is used for TTT both in concept space and in CLIP reconstructions. Unless otherwise specified, Adam's parameters follow the default values in PyTorch (Paszke et al., 2019).

## F.4   SCALING EXPERIMENTS ON MNIST

In the scaling experiments reported in Section 4, we trained multiple LeNet (LeCun et al., 1998a) convolutional neural networks (CNNs) at different model scales. The architecture comprises two convolutional layers, each with ReLU activation and $2 \times 2$ max-pooling, followed by a fully connected classification head. The reference models of varying sizes were trained with the hyperparameters listed in Table 4. Optimization was performed with Adam (Kingma & Ba, 2015).

For the model scaling plot in Figure 4, we trained each reference model across five random seeds and applied both TTT and majority voting. For TTT, all model parameters were frozen except for

| | Model parameters | | | | | | | | | | | |
|---|---|---|---|---|---|---|---|---|---|---|---|---|
| | **280** | **600** | **1268** | **1976** | **2985** | **4430** | **6386** | **11770** | **24294** | **40818** | **61342** | **85866** |
| Learning rate | 6e-3 | 2e-3 | 1e-3 | 8e-4 | 6e-4 | 6e-4 | 2e-3 | 2e-3 | 2e-3 | 6e-4 | 2e-3 | 2e-3 |
| Batch Size | 500 | 200 | 400 | 600 | 300 | 300 | 400 | 300 | 300 | 100 | 300 | 500 |
| Epochs | 50 | 50 | 100 | 100 | 100 | 100 | 100 | 100 | 100 | 50 | 100 | 100 |

Table 4: Hyperparameters for globally trained CNNs.

the final linear layer, which was fine-tuned from its pre-trained initialization using the hyperparameters in Table 5. Majority voting was performed with neighborhood sizes specified in Table 6. All hyperparameters were obtained from a hyperparameter optimization on the validation set. As the experiments were repeated over five seeds, we used 200 bootstrap iterations per seed to compute confidence intervals.

| | Model parameters | | | | | | | | | | | |
|---|---|---|---|---|---|---|---|---|---|---|---|---|
| | **280** | **600** | **1268** | **1976** | **2985** | **4430** | **6386** | **11770** | **24294** | **40818** | **61342** | **85866** |
| Learning rate | 0.05 | 5e-3 | 0.01 | 0.01 | 5e-3 | 5e-3 | 1e-3 | 5e-3 | 1e-3 | 1e-3 | 5e-3 | 1e-3 |
| Epochs | 50 | 200 | 200 | 200 | 200 | 200 | 200 | 200 | 200 | 200 | 200 | 100 |
| Neighbors | 200 | 50 | 10 | 200 | 200 | 200 | 50 | 100 | 50 | 200 | 300 | 100 |

Table 5: Hyperparameters for TTT on the linear head of the CNNs.

| | Model parameters | | | | | | | | | | | |
|---|---|---|---|---|---|---|---|---|---|---|---|---|
| | **280** | **600** | **1268** | **1976** | **2985** | **4430** | **6386** | **11770** | **24294** | **40818** | **61342** | **85866** |
| Neighbors | 8 | 5 | 8 | 5 | 3 | 4 | 6 | 4 | 4 | 4 | 5 | 5 |

Table 6: Hyperparameters for majority voting based on last-hidden-layer features of the CNNs.

To generate the dataset scaling plot in Figure 5, we randomly subsampled the training set while explicitly ensuring a uniform distribution across the 10 class labels. Here, TTT was performed by minimizing the cross-entropy loss over the $k = 80$ nearest neighbors. Optimization was performed using Adam with a learning rate of 0.02 for 500 epochs.

### F.5 CONNECTION TO MoEs

Given an underparameterized global model, a natural alternative to specializing to each test-time task as in TTT is to instead specialize individual "expert" models to subsets of tasks, routing test-time tasks to few of these experts. Such mixture of experts (MoEs; Shazeer et al., 2017; Fedus et al., 2022; Bertolissi et al., 2025) have been shown to be an effective architecture for foundation models (e.g., Dai et al., 2024).

To see whether our findings extend to MoEs, we train multiple experts (each being a different linear head) based on the MoE architecture of Bertolissi et al. (2025), and evaluate accuracy as we scale the number of experts. We find that a larger number of experts increases the capacity of the model and improves accuracy, highlighting that MoEs are a promising approach to specialization without increasing inference cost.

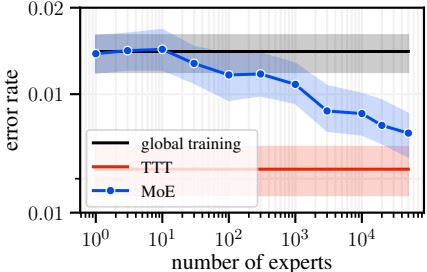

Figure 9: Classification error on MNIST. We compare the global classifier, TTT, and the MoE. Increasing the number of experts allows MoE to approach TTT performance with inference cost comparable to the global model.

We used a pre-trained CNN to extract last-layer embeddings from MNIST images. Following Bertolissi et al. (2025), for a given number of experts, each expert was associated with a cluster

| | Number of experts | | | | | | | | | | |
|---|---|---|---|---|---|---|---|---|---|---|---|
| | 1 | 3 | 10 | 30 | 100 | 300 | 1e3 | 3e3 | 10e3 | 20e3 | 50e3 |
| Learning rate | 6e-4 | 2e-4 | 6e-4 | 1e-3 | 8e-4 | 4e-4 | 6e-4 | 4e-4 | 6e-4 | 4e-4 | 4e-4 |
| Epochs | 2 | 1 | 2 | 1 | 2 | 3 | 10 | 30 | 20 | 50 | 40 |
| Neighbors | 60 | 50 | 30 | 60 | 30 | 40 | 30 | 20 | 30 | 30 | 20 |

Table 7: Hyperparameters for the mixture of experts (MoE) model based on last-hidden-layer features of the CNN.

centroid obtained by partitioning the training set into clusters via $k$-means. Expert training was performed by fine-tuning the pre-trained linear head on the nearest neighbors of the assigned centroid in embedding space. The fine-tuning hyperparameters are provided in Table 7, obtained by tuning on the validation set. At test time, inputs were first mapped through the CNN encoder, after which the single closest cluster centroid was selected based on $L_2$-distance. The corresponding expert head then produced the final prediction.

### F.6 Scaling experiments on ImageNet

In our scaling experiments on ImageNet, we systematically vary the dimensionality and architecture of downstream classifiers, enabling a thorough analysis of scalability and representation quality. The following subsections detail our embedding extraction procedure, the dataset partitioning scheme, and the training protocols used for all baselines and scaling methods.

**Embedding Extraction.** For the ImageNet experiments, we use image embeddings derived from the CLIP vision-language model (Radford et al., 2021). Specifically, we employ the ViT-B/32 variant of CLIP, as provided by the HuggingFace Transformers library. Images are first preprocessed by applying resizing, center cropping, and pixel normalization to match CLIP's training setup. The preprocessed images are then passed through the CLIP vision encoder, yielding 512-dimensional representation vectors. To ensure stability and scale invariance, we normalize each embedding to unit length using the $L_2$ norm.

**Dataset Splits.** Since ImageNet's official test labels are held out, we report results on the official validation set, treating it as our test set. For training purposes, we partitioned the 1.28M-image training set into a reduced training set and an artificial validation set. The artificial validation set was constructed by stratified sampling 50000 images to ensure a balanced class distribution, giving it the same size as the official validation set. The remaining images were used for training. Unless otherwise stated, all reported results are based on evaluation on the official validation set.

**Baseline Linear Classifier.** As a first baseline, we trained a linear classifier on the 512-dimensional normalized CLIP embeddings. The linear head is a fully connected layer mapping the embeddings to 1000 output classes, corresponding to the ImageNet labels. Training was performed with the Adam optimizer at a learning rate of 0.001, using a batch size of 250 for 50 epochs. The model was trained with standard categorical cross-entropy loss, without additional regularization, and achieved a test accuracy of 78.33 %.

**TTT for Base Model.** Building upon the baseline linear classifier, we apply a test-time training procedure to adapt the model at inference-time. Specifically, we fine-tune this linear head for each test sample by minimizing the cross-entropy loss over the set of $k = 600$ nearest neighbors retrieved in the original CLIP embedding space. By optimizing the error rate on the validation set, we determined that using $k = 600$ neighbors with a learning rate of 0.02, a batch size equal to the number of neighbors, and 50 training epochs, combined with the Adam optimizer, perform near-optimally across all model scales. Consequently, we adopted this universal set of hyperparameters for the ImageNet experiments.

**Two-Layer MLP Projections.** To assess the impact of embedding dimensionality, we trained two-layer multi-layer perceptrons (MLPs) as classification heads on top of the CLIP embeddings. The first hidden layer had variable size, ranging from 250 to 4000 neurons, followed by a ReLU activation. The second layer maps the hidden representation to the 1000 ImageNet classes. Dropout is

applied before the final layer to mitigate overfitting. Training was conducted with the Adam optimizer, using the hyperparameters specified in Table 8, which were selected via optimization on the validation set.

| | Model parameters | | | | | | | | | |
|---|---|---|---|---|---|---|---|---|---|---|
| | 3.8e5 | 7.6e5 | 1.1e6 | 1.5e6 | 1.9e6 | 2.3e6 | 3.0e6 | 3.8e6 | 4.5e6 | 6.1e6 |
| Hidden dimension | 250 | 500 | 750 | 1000 | 1250 | 1500 | 2000 | 2500 | 3000 | 4000 |
| Learning Rate | 4.0e-4 | 3.5e-4 | 3.0e-4 | 4.0e-4 | 3.5e-4 | 4.0e-4 | 4.0e-4 | 3.5e-4 | 4.5e-4 | 4.5e-4 |
| Weight Decay | 0 | 0 | 0 | 0 | 0 | 0 | 0 | 0 | 0 | 0 |
| Batch Size | 450 | 350 | 300 | 450 | 350 | 300 | 400 | 450 | 450 | 650 |
| Num Epochs | 50 | 50 | 50 | 50 | 50 | 50 | 50 | 50 | 50 | 50 |
| Dropout Rate | 0.05 | 0.25 | 0.3 | 0.35 | 0.45 | 0.55 | 0.6 | 0.65 | 0.7 | 0.7 |

Table 8: Hyperparameters for globally training the MLP heads across different model sizes.

After training, we freeze the first hidden layer and project the original 512-dimensional embeddings into this hidden space. The resulting hidden representations, taken after the ReLU activation, serve as our scaled embeddings for test-time training. Specifically, we fine-tune the pre-trained linear MLP head on the set of $k$ nearest neighbors by minimizing the cross-entropy loss using Adam. This setup allows us to systematically examine how performance and representation quality vary with embedding dimensionality. The hyperparameters used for this procedure were obtained via hyperparameter optimization on the validation set, and we found that this universal set is nearly optimal across all model sizes, as reported in Table 9. Loss optimization was performed in a full-batch setting, with the batch size equal to the number of neighbors.

| Hyperparameter | Value |
|---|---|
| Number of neighbors | 100 |
| Learning rate | 5e-3 |
| Batch size | 100 |
| Epochs | 50 |

Table 9: Hyperparameters for TTT on the linear head of the MLPs.

**Majority Voting.** As an alternative baseline, we leverage the learned feature spaces of the two-layer MLPs and apply a simple majority voting protocol based on nearest neighbors. Specifically, we first map the original 512-dimensional CLIP embeddings into the MLP's hidden feature space and then identify its $k = 10$ nearest neighbors for any given test sample. Our empirical analysis showed that this neighborhood size performs well across all model scales. The predicted class is assigned as the most frequent (majority) class label among these neighbors. This approach parallels the neighbor selection used in test-time training (TTT) but replaces fine-tuning with a straightforward plurality vote. Majority voting thus serves as a simple, non-parametric baseline to assess the quality of the scaled embeddings, providing insight into the clustering and class separability properties of the learned feature space.

### F.7 SCALING EXPERIMENTS FOR LANGUAGE MODELING

We use the open-source implementation[10] of Hardt & Sun (2024) and evaluate the Qwen2.5 family of base models (Qwen et al., 2025). We summarize hyperparameters in Table 10. During test-time training, we perform gradient steps sequentially for each sequence in the neighborhood. Sequences in the neighborhood that exceed the maximum sequence length are split into chunks of the maximum length. This means that a single neighbor can result in multiple gradient steps during TTT.

---

[10]https://github.com/socialfoundations/tttlm

| Hyperparameter | Value |
|---|---|
| Number of neighbors | 50 |
| Learning rate | 2e-4 |
| Adam's $\epsilon$-value | 1e-8 |
| Max. sequence length in tokens | 1024 |
| LoRA rank | 64 |

Table 10: Hyperparameters for language modeling on the Pile.

