# OpenReview forum: "Specialization after Generalization: Towards Understanding Test-Time Training in Foundation Models"
_ICLR.cc/2026/Conference — ICLR 2026 Poster_

### Official Review · Reviewer_V2Zv · 2025-10-28

**Soundness:** 3
**Presentation:** 3
**Contribution:** 3
**Rating:** 6
**Confidence:** 2

**Summary:**

The authors analyze a model of test time training and argue that test-time training can provide benefits even in-distribution, so long as the model is capacity constrained.

**Strengths:**

- The paper is well-written and easy to follow
- The observation that TTT can help performance even for in-distribution inputs, and that this benefit is more pronounced in the underparameterized regime, is interesting, and to my knowledge, novel
    - The authors show how the effect of TTT may be to 'disentangle' representations in an underparameterized representation space
- The authors confirm their predictions with experiments across scales & domains: from MNIST to language models

**Weaknesses:**

- The authors consider TTT only in the regime where the last linear layer is retrained. While this does ensure that the representations learned during pretraining remain unchanged, it also puts a capacity constraint on the TTT model. Do the observations in the paper still hold up if we train more of the model at test time?
    - I am particularly concerned that the phenomena in the paper may be a special case induced by the linearity of the final layer. I think even an experiment where TTT modifies the last 2 layers would go some way to mitigating this concern.
- It is not clear to me where the under/overparameterized regimes begin and end. Could the authors note this on the figures?
- In Figure 4, at least for the MNIST dataset, if the authors' hypothesis is correct, it should be possible to push to a regime where TTT yields no improvement. Is it feasible to extend the x-axis further, at least for the MNIST models?

**Questions:**

- In the theoretical framing, the relevant dimensionality seems to be the dimension of the latent space (compared to the number of "concepts" needed to solve the task). Despite this, in several places (e.g. Fig 4), the *total* number of parameters is used as a proxy for this dimensionality. Are there experiments you could run to disentangle these two different notions of dimensionality? For example, changing width/depth of MNIST models independently, do you see different behavior?
- Can you clearly delineate the under/overparameterized regimes on Figs 4,5? It is not clear to me how much training data was used in each setting.
- Is there a way to show that during TTT, the "effective" dataset size (during TTT we allow the model to perform poorly on data not in the current test set, so the effective dataset is small) becomes small enough that the model is again ~overparameterized? Can you connect this to double descent, at least in the MNIST setting?
    - I am speaking loosely here, but the idea is that doing TTT in-distribution amounts roughly to retraining the model on a smaller dataset, which we know from the double descent literature has the potential to *improve* performance.

---

> ### Author Response · Authors · 2025-11-22
>
> Thank you for your review and detailed comments! We are happy to address each of the points individually.
>
> > The paper is well-written and easy to follow
> The observation that TTT can help performance even for in-distribution inputs, and that this benefit is more pronounced in the underparameterized regime, is interesting, and to my knowledge, novel.
> The authors confirm their predictions with experiments across scales & domains: from MNIST to language models
>
> Thank you!
>
> > The authors consider TTT only in the regime where the last linear layer is retrained. While this does ensure that the representations learned during pretraining remain unchanged, it also puts a capacity constraint on the TTT model. Do the observations in the paper still hold up if we train more of the model at test time?
> I am particularly concerned that the phenomena in the paper may be a special case induced by the linearity of the final layer. I think even an experiment where TTT modifies the last 2 layers would go some way to mitigating this concern.
>
> Thank you for this thoughtful suggestion. You are entirely correct: while last-layer TTT imposes a potential capacity constraint, it guarantees that the rich semantic structure in the feature space that is learned during pre-training is not distorted. This fixed-feature regime was chosen for the MNIST and ImageNet experiments to closely align with our theoretical model.
>
> To address the concern that our results might be artifacts of the last layer’s capacity constraints, we conducted an additional experiment performing full end-to-end fine-tuning (updating all parameters) while using the last layer for neighbor selection. The results (see Figure 7 in Appendix D) show that the error rates and qualitative trends remained nearly identical between the last-layer-only and the fully end-to-end approaches.
>
> > In Figure 4, at least for the MNIST dataset, if the authors' hypothesis is correct, it should be possible to push to a regime where TTT yields no improvement. Is it feasible to extend the x-axis further, at least for the MNIST models?
>
> You are raising an important point: our theoretical model implies that TTT improves performance by compensating for the underparameterization of the global model. Therefore, for sufficiently large (overparameterized) models, the improvements from TTT should vanish. To validate this hypothesis, we extended the x-axis for the MNIST experiments as you suggest. As shown in the revised Figure 4 (left), we observe exactly the predicted behavior: up to approximately 10^4 parameters, TTT consistently improves performance. As we scale the models further, the TTT gain diminishes to zero, converging to the globally trained baseline.
>
> > It is not clear to me where the under/overparameterized regimes begin and end. Could the authors note this on the figures?
>
> We agree with you that visualizing these regimes could aid interpretation. However, pinpointing the exact theoretical boundary between under- and overparameterization on real-world data (like MNIST) is non-trivial, as it depends on the effective dimensionality of the data manifold rather than just sample count.
>
> In the revised MNIST plot, we observe that TTT error rates converge to the global baseline for the largest model shown. Intuitively, the grid line at around 10^4 parameters can be seen as approximately indicating the transition from the capacity-constrained regime (underparameterized) to the overparameterized regime.
>
> > In the theoretical framing, the relevant dimensionality seems to be the dimension of the latent space (compared to the number of "concepts" needed to solve the task). Despite this, in several places (e.g. Fig 4), the total number of parameters is used as a proxy for this dimensionality. Are there experiments you could run to disentangle these two different notions of dimensionality? For example, changing width/depth of MNIST models independently, do you see different behavior?
>
> We agree that while total parameter count is a useful proxy, the feature dimension is the theoretically relevant quantity for our analysis.
>
> To explicitly disentangle these notions as suggested, we have added a complementary analysis to the Appendix D, Figure 6. In this experiment, we investigate the scaling behavior specifically as a function of network width (= feature dimension), rather than total parameters. The results demonstrate that scaling the feature dimension yields the same qualitative and quantitative behavior as the parameter-based scaling shown in the main body. Specifically, TTT consistently outperforms global training and majority vote, with the performance gap narrowing as the hidden dimension increases.
>
> > It is not clear to me how much training data was used in each setting.
>
> In the model scaling experiments (Figure 4), we use the full training data for each model scale.

---

> > ### Author Response · Authors · 2025-11-22
> >
> > > Is there a way to show that during TTT, the "effective" dataset size (during TTT we allow the model to perform poorly on data not in the current test set, so the effective dataset is small) becomes small enough that the model is again ~overparameterized? Can you connect this to double descent, at least in the MNIST setting?
> > I am speaking loosely here, but the idea is that doing TTT in-distribution amounts roughly to retraining the model on a smaller dataset, which we know from the double descent literature has the potential to improve performance.
> >
> > Thank you for highlighting this compelling connection! The results in Table 2 provide useful empirical guidance.
> >
> > * For the MNIST experiment in particular, we observe strong evidence that this setting operates in the overparameterized / interpolation regime. The TTT head achieves 100% accuracy on the local neighborhood (driving loss to zero) but suffers significantly on the global test set. This indicates the model has sufficient capacity to fully interpolate the local noise while diverging from the pre-trained global solution.
> > * For the ImageNet experiment, the picture looks different. First, the neighborhood accuracy is high (95%), but it does not reach perfect interpolation (100%). Second, and more surprisingly, the model retains nearly all its global performance (dropping only ~1% on the full test set). This indicates that the TTT update to the head is minimal.
> >
> > Taken together, these observations suggest that TTT on MNIST operates in an overparameterized regime, whereas TTT on ImageNet may still lie in an underparameterized regime.
> >
> > The new end-to-end TTT experiment (see Figure 7 in Appendix D) provides the clearest connection to your hypothesis. Here, we adapt all model weights on the small test batch. This creates a scenario where the ratio of parameters to effective data clearly becomes extremely large, pushing the estimator far into the overparameterized regime. Consistent with the "benign overfitting" [1] observed in the right-hand side of the double descent curve, optimization becomes easier: the model finds a minimum norm solution that fits the test features, thereby reducing the variance caused by distribution shifts. While MacKay [2] framed this as fitting a simpler local estimator, in modern deep learning terms, it corresponds to exploiting the variance reduction of extreme overparameterization to flatten the risk curve.
> >
> > ---
> >
> > Thank you again for your detailed review. Please let us know if you have any remaining questions or concerns.
> >
> > [1] Bartlett et al. Benign Overfitting in Linear Regression. arXiv:1906.11300 2019
> >
> > [2] MacKay. A Practical Bayesian Framework for Backpropagation Networks. Neural Comput. 1992

---

### Official Review · Reviewer_v2NY · 2025-10-31

**Soundness:** 3
**Presentation:** 4
**Contribution:** 3
**Rating:** 6
**Confidence:** 3

**Summary:**

The paper attempts to answer why and when does test-time training (TTT) help foundation models even when test data are in-distribution? It assumes that foundation models are under-parameterized, and that TTT provides "specialization after generalization". Behind this is the Linear Representation Hypothesis (LRH): high-level "concepts" are sparse and linearly represented but superimposed onto a lower-dimensional feature space.

The authors reach 3 key observations: (O1) neighborhoods in learned features preserve the geometry from the concept space; (O2) each test neighborhood only depends on a few active concepts; (O3) standard TTT implicitly finds sparse solutions in concept space. Usinng sparse autoencoders and scaling studies on MNIST, ImageNet (with CLIP features), and The Pile (with LoRA), they show that TTT improves accuracy especially when models are small and under-parameterized, and that the benefit shrinks as models scale up in terms of number of parameters. Overall, the paper provides evidence that TTT’s success comes from reallocating model capacity locally.

**Strengths:**

Comprehensive and helpful literature review. Text is well-written and structured. Figures are clear and support main claims.

The three observations (O1--O3) are backed up with both experiments and analysis.

The math connects with the experimental results and explains why global training can fail when features are mixed.

Understanding when TTT helps (even with no distribution shift) is relevant for practical deployment of large models. The paper offers practical guidelines for scaling, such as how to choose neighborhood size and when TTT is likely to be worth using.

**Weaknesses:**

Several results depend on strong ideas like sparse concept spaces and preserved neighborhood geometry. It would help to test these assumptions on different architectures or layers to show they hold more generally.

The LoRA test-time training uses only one step and on a small subset (1%), 50 neighbors. Running more steps, larger splits, or other LLMs could show whether the findings scale.

The tests on images (LeNet-5 and pre-trained CLIP ViT) only update the last layer. Adding results with full end-to-end fine-tuning would make the findings more convincing.

Although the paper does a good job at listing all its main assumptions, it mostly deals with them by giving arguments for why those assumptions might be reasonable. But the paper is lacking on discussing what could happen in scenarios where they do not hold. For example, TTT assumes neighborhoods correspond to a single uniform local concept. The authors should discuss what happens when that assumption breaks (e.g. due to label noise, or poor neighborhood size selection), since specialization in that case may lead to worse generalization.

**Questions:**

In the LoRA setup, the paper uses one gradient step. How was this chosen? Were multiple step sizes or LoRA ranks tested? Would they improve results?

In Appendix E.4, Table 4, different values are reported for the number of neighbors used for TTT on MNIST, depending on the model size. How were these chosen? In contrast, why was a fixed value of 100 used for ImageNet and of 50 for language modeling?

---

> ### Author Response · Authors · 2025-11-22
>
> Thank you for your review and detailed comments and questions! We are happy to address each of the points individually.
>
> > Comprehensive and helpful literature review. Text is well-written and structured. Figures are clear and support main claims.
> The three observations (O1--O3) are backed up with both experiments and analysis.
> The math connects with the experimental results and explains why global training can fail when features are mixed.
> Understanding when TTT helps (even with no distribution shift) is relevant for practical deployment of large models. The paper offers practical guidelines for scaling, such as how to choose neighborhood size and when TTT is likely to be worth using.
>
> Thank you!
>
> > Several results depend on strong ideas like sparse concept spaces and preserved neighborhood geometry. It would help to test these assumptions on different architectures or layers to show they hold more generally.
>
> Thank you for highlighting this. Our results indeed are based on the presence of sparse concept spaces and locally consistent neighborhood geometry. Importantly, both assumptions are strongly supported by a growing body of empirical evidence on the structure of deep representations, which spans multiple architectures, modalities, and scales.
>
> * **Sparse concept spaces**: Recent progress on sparse autoencoders (SAEs) shows that models implicitly encode information in sparse, highly interpretable feature spaces. Large-scale empirical studies [1, 2] have demonstrated that sparse, interpretable features can be recovered from dense representations across a wide variety of architectures (including Transformers and CNNs) and scales. This suggests that sparsity is not an artifact of a specific model, but an emergent property of overparameterized networks solving high-dimensional tasks.
> * **Preserved neighborhood geometry**: Likewise, the linear representation hypothesis posits that semantic concepts correspond to approximately linear directions in activation space, which inherently preserves neighborhood geometry (allowing for vector arithmetic). This has been validated across highly diverse domains, including sentiment [3], the state of a game board [4], and representations of space and time in LLMs [5].
>
> Given this extensive prior validation, we believe these properties are general enough to support our claims.
>
> > The LoRA test-time training uses only one step and on a small subset (1%), 50 neighbors. Running more steps, larger splits, or other LLMs could show whether the findings scale.
> In the LoRA setup, the paper uses one gradient step. How was this chosen? Were multiple step sizes or LoRA ranks tested? Would they improve results?
>
> For our LLM experiments, we adopt the implementation of [6], previously also analyzed by [7]. The LoRA training uses batch size 1, i.e., one step per each of the 50 neighbors (with the caveat that sequences longer than the maximum sequence length are split into multiple individual sequences / gradient steps). The previous works of [6,7] provide extensive analyses with varying models, hyperparameters, and across numbers of steps. We used the same LoRA rank (64) as reported by [7]. Our work extends the previous work by evaluating across models of varying scale from the same architecture.
>
> > The tests on images (LeNet-5 and pre-trained CLIP ViT) only update the last layer. Adding results with full end-to-end fine-tuning would make the findings more convincing.
>
> Thank you for this great suggestion. While our initial setup froze the feature extractor to strictly align with our theoretical model (which assumes a fixed representation space), we agree that validating this with full fine-tuning is important.
>
> To address this, we conducted an additional experiment performing end-to-end (E2E) fine-tuning. Specifically, we used the last hidden layer to identify neighbors (as in the original method) but allowed gradients to update all model parameters. The results (see Fig. 7 in Appendix D) show that E2E TTT yields nearly identical error rates to last-layer TTT on MNIST. This indicates that the qualitative phenomena and the theoretical alignment persist even when the feature extractor is adaptive. Moreover, our LLM experiments (and prior work in this domain) already leverage more expressive training setups, observing the same behavioral trends.

---

> > ### Author Response · Authors · 2025-11-22
> >
> > > Although the paper does a good job at listing all its main assumptions, it mostly deals with them by giving arguments for why those assumptions might be reasonable. But the paper is lacking on discussing what could happen in scenarios where they do not hold. For example, TTT assumes neighborhoods correspond to a single uniform local concept. The authors should discuss what happens when that assumption breaks (e.g. due to label noise, or poor neighborhood size selection), since specialization in that case may lead to worse generalization.
> >
> > Thank you for this feedback and for acknowledging that our paper clearly lists its main assumptions. We agree that discussing potential failure modes is essential. Below, we summarize the key points:
> >
> > * **Local concepts**: TTT does not require each neighborhood to correspond to a single concept. Instead, we assume local simplicity, meaning that the neighborhood distribution can be largely explained by a few dominant concepts. If this assumption fails, TTT cannot effectively specialize, as the local gradient signal becomes ambiguous.
> > * **Label noise**: High levels of label noise affect both the global pre-trained model and the TTT adaptation. While the global model struggles due to noisy decision boundaries, TTT can overfit to noisy samples during adaptation. Extreme label noise remains a fundamental limitation for any learning method.
> > * **Neighborhood size selection**: As illustrated in Fig. 5, selecting neighborhood size involves a trade-off between statistical variance and concept relevance. Small neighborhoods lead to high variance due to limited samples, while large neighborhoods may include irrelevant or distant concepts, violating local simplicity and introducing bias. The optimal neighborhood size balances sufficient statistics with semantic consistency to enable effective specialization.
> >
> > Generally speaking, while each of the hypotheses 1-3 in Section 5.1 are grounded in our empirical observations from Section 3, TTT might fail to improve accuracy if any of the hypotheses were to be violated.
> > We have clarified this in the revised Section 5.
> >
> > > In Appendix E.4, Table 4, different values are reported for the number of neighbors used for TTT on MNIST, depending on the model size. How were these chosen? In contrast, why was a fixed value of 100 used for ImageNet and of 50 for language modeling?
> >
> > Thank you for this remark. The hyperparameters reported in Appendix E.4, Table 4 were obtained by tuning on the validation set. The same procedure was applied for the hyperparameters in Tables 3 and 5.
> >
> > For the ImageNet experiments, we fixed the number of neighbors to 100, as the hyperparameter search indicated that this choice was close to optimal across all hidden dimensions. In contrast, the MNIST classifier exhibits a higher sensitivity to this parameter. We attribute this to the low intrinsic dimensionality and highly clustered structure of the MNIST data. As a result, small changes in the neighborhood size can include samples from different classes or stroke styles, leading to discontinuous shifts in local statistics and consequently in the adaptation dynamics.
> >
> > By comparison, the CLIP representations used for ImageNet (and the embeddings of the language model) inhabit a high-dimensional, semantically smooth space. In such representations, local neighborhoods are more homogeneous, so varying the number of neighbors only weakly affects local averages and gradients.
> >
> > To clarify how the neighborhood hyperparameters were selected, we have added remarks to the corresponding paragraphs in the appendix.
> >
> > ---
> >
> > Thank you again for your detailed review. Please let us know in case you have any remaining questions or concerns.
> >
> > [1] Bricken et al. Towards Monosemanticity: Decomposing Language Models With Dictionary Learning, Transformer Circuits Thread, 2023.
> >
> > [2] Cunningham et al. Sparse Autoencoders Find Highly Interpretable Features in Language Models. arXiv:2309.08600 2023
> >
> > [3] Tigges et al. Linear representations of sentiment in large language models. arXiv:2310.15154, 2023.
> >
> > [4] Nanda et al.. Emergent linear representations in world models of self-supervised sequence models. BlackboxNLP, 2023.
> >
> > [5] Gurnee and Tegmark. Language models represent space and time. ICLR, 2024.
> >
> > [6] Hardt & Sun. Test-time training on nearest neighbors for large language models. In ICLR, 2024.
> >
> > [7] Hübotter et al. Efficiently learning at test-time: Active fine-tuning of LLMs. In ICLR, 2025.

---

> > > ### Comment · Reviewer_v2NY · 2025-11-23
> > >
> > > I believe this authors have sufficiently addressed my main concerns, both by providing relevant additional references and by running additional experiments. I have updated my score to reflect this.

---

> > > > ### Author Response · Authors · 2025-11-26
> > > >
> > > > Thank you for the feedback and for raising your score. We appreciate the time you took to review our rebuttal!

---

### Official Review · Reviewer_1Gcj · 2025-11-01

**Soundness:** 3
**Presentation:** 3
**Contribution:** 3
**Rating:** 8
**Confidence:** 1

**Summary:**

This paper investigates test-time training (TTT) and proposes that its effectiveness arises from enabling specialization after generalization, where models locally adapt to test data. Using the linear representation hypothesis, the authors model how TTT reweights a small subset of relevant concepts in an underparameterized feature space, achieving lower in-distribution error than global training. Empirical and theoretical results across vision and language tasks show that TTT offers the greatest gains for underparameterized models, with benefits diminishing as model capacity increases.

**Strengths:**

1. The paper introduces a clear and formal connection between TTT and the linear representation hypothesis (LRH), providing an interpretable mathematical model for understanding why and when TTT improves performance.
2. The use of SAEs to approximate the latent “concept space” offers a convincing and interpretable lens to study TTT’s behavior, even for readers (like myself) who are not deeply specialized in TTT.
3. Experiments on both image (ImageNet, MNIST) and language (Pile) domains systematically support the theoretical claims, demonstrating consistent trends across scales and tasks.

**Weaknesses:**

As someone not specialized in test-time training, I found the exposition somewhat difficult to follow—the paper would benefit from a clearer introductory section that outlines core TTT mechanisms and variants before diving into the theoretical framing.

**Questions:**

Could the author elaborate on the definition of test-time training?

---

> ### Author Response · Authors · 2025-11-22
>
> Thank you for your review, and the positive evaluation of our work. We are happy to address each of the points individually.
>
> > The paper introduces a clear and formal connection between TTT and the linear representation hypothesis (LRH), providing an interpretable mathematical model for understanding why and when TTT improves performance.
>
> > The use of SAEs to approximate the latent “concept space” offers a convincing and interpretable lens to study TTT’s behavior, even for readers (like myself) who are not deeply specialized in TTT.
>
> > Experiments on both image (ImageNet, MNIST) and language (Pile) domains systematically support the theoretical claims, demonstrating consistent trends across scales and tasks.
>
> Thank you for this positive assessment.
>
> > As someone not specialized in test-time training, I found the exposition somewhat difficult to follow—the paper would benefit from a clearer introductory section that outlines core TTT mechanisms and variants before diving into the theoretical framing.
>
> Thank you for highlighting this important consideration! We want our findings and theoretical results to be accessible to the broader community, not only specialists in TTT, so we very much appreciate this perspective. Due to strict space constraints in the submission, we condensed the background on Test-Time Training to prioritize the theoretical and empirical results.
>
> The while the literature on test-time training is still relatively new and emerging there are several themes that have been pursued in prior work:
>
> * **TTT in local neighborhoods**: Works that perform test-time training on “similar” data provided at test-time [e.g., 1,2,3]. This is closest to our work.
> * **TTT as a recurrent architecture for sequence modeling**: A family of recent works explore TTT as a recurrent architecture for sequence modeling that addresses the memory bottleneck of transformers [4,5]. The above work [1,2,3] can be seen as simplified instances of this setting where the full model is considered as memory and the memory is updated in infrequent chunks (not after every token).
> * **Training on the test-task without explicit supervision**: Several works train on the test task without explicit labels, e.g., by minimizing entropy of the predictions [e.g., 6,7,8]. Extending our results to a setting where we learn without a labeled neighborhood is an exciting direction for future work.
>
> The unifying element of these works is that the model is trained at test-time for a particular prediction. This has been demonstrated empirically to improve predictions under distribution-shift. This work provides a systematic study of a simple TTT method, showing that it also improves predictions in-distribution using only already-seen data.
>
> ---
>
> Thank you again for your review. Please do not hesitate to reach out in case you have any remaining questions or concerns.
>
> [1] Hardt & Sun. Test-time training on nearest neighbors for large language models. In ICLR, 2024.
>
> [2] Hübotter et al. Efficiently learning at test-time: Active fine-tuning of LLMs. In ICLR, 2025.
>
> [3] Akyürek et al.  The surprising effectiveness of test-time training for few-shot learning. In ICML, 2025.
>
> [4] Sun et al. Learning to (learn at test time): Rnns with expressive hidden states. In ICML, 2025.
>
> [5] Zhang et al. Test-time training done right. arXiv preprint arXiv:2505.23884, 2025.
>
> [6] Wang et al. Tent: Fully test-time adaptation by entropy minimization. ICLR, 2021.
>
> [7] Zhang et al. Memo: Test time robustness via adaptation and augmentation. In NeurIPS, 2022.
>
> [8] Zuo et al. Ttrl: Test-time reinforcement learning. In NeurIPS, 2025.

---

### Official Review · Reviewer_zPcj · 2025-11-01

**Soundness:** 4
**Presentation:** 4
**Contribution:** 4
**Rating:** 8
**Confidence:** 2

**Summary:**

The paper studies the mechanisms of TTT in Foundation Models and its effectiveness in different contexts (the model capacity or dataset complexity). The work asks an important question of the TTT's effectiveness performed on the already seen data samples, which lets us focus on the process itself and not on the data. The claims made by the authors are clearly stated and supported by a thorough experimental section spanning experiments from image classification to NLP.

**Strengths:**

1. The work focuses on the extremely important topic, as the TTT plays an important role in the current cutting-edge research after showing its effectiveness in different domains; thus, I value the significance of the results highly.
2. The idea to isolate the effect of TTT from the data (i.e. to study the effectiveness of TTT on already seen data) makes the analysis much more convincing to me. Still, I have to point out that I'm only vaguely familiar with this field, so my novelty judgment can be off.
3. Both the presentation and the experimental design are thoughtful and help the reader to understand the main claims made by the authors, making it a solid piece of work with carefully balanced claims.

**Weaknesses:**

I did not find any important weaknesses in the paper. I have some questions which I will ask in the next section.

**Questions:**

1. Can we conclude that current LLMs, given their performance gains after TTT, are underparameterized models relative to their datasets? If so, why do the authors think that scaling the LLMs to even greater capacities ceased to be the main theme in LLM training, and we started to observe the diminished returns from scaling?


2. How important is the metric used for generating the neighbourhood? The authors studied L2 distance, which was equivalent to cosine similarity, but there are multiple different ways to define the neighbourhood. Can we expect this metric to be, on average, the best approach, or would this choice be important when applying the approach to different datasets, modalities or architectures?

3. Line 239-241: The Authors claim that the sparse TTT classifier is functionally equivalent based on 89% match (except for pathological cases). Could the authors elaborate on what "pathological" means? 11% seems to be relatively high and I don't understand where should I expect these two would deviate.

---

> ### Author Response · Authors · 2025-11-22
>
> We would like to thank you for the positive evaluation of our submission, and the detailed and clear feedback. We will address each point individually.
>
> > The work focuses on the extremely important topic, as the TTT plays an important role in the current cutting-edge research after showing its effectiveness in different domains; thus, I value the significance of the results highly.
>
> Thank you for recognizing the importance of this topic and the significance of our results.
>
> > Can we conclude that current LLMs, given their performance gains after TTT, are underparameterized models relative to their datasets? If so, why do the authors think that scaling the LLMs to even greater capacities ceased to be the main theme in LLM training, and we started to observe the diminished returns from scaling?
>
> Thank you for this question, which highlights a fundamental aspect of LLM scaling. Our position is that contemporary foundation models remain underparameterized relative to the information-theoretic complexity of their training corpus. They act as lossy compressors of their data [1] and therefore cannot perfectly memorize or represent the full distribution [2].
>
> Empirical scaling laws follow a power-law trajectory, implying that a linear reduction in error necessitates orders-of-magnitude increases in model scale. Intuitively, this creates a regime of rapidly diminishing returns: as the model captures the core data distribution, resolving the remaining long-tail ambiguity becomes disproportionately expensive. Consequently, further static scaling eventually becomes economically intractable. In this work, we analyze Test-Time Training (TTT) as a more efficient alternative that bypasses this saturation by allocating inference compute to dynamically adapt the model to the specific input, rather than expanding global capacity.
>
> > How important is the metric used for generating the neighbourhood? The authors studied L2 distance, which was equivalent to cosine similarity, but there are multiple different ways to define the neighbourhood. Can we expect this metric to be, on average, the best approach, or would this choice be important when applying the approach to different datasets, modalities or architectures?
>
> This is a great question! The choice of metric for defining the neighborhood does indeed influence performance, yet exploring alternative metrics is not the main focus of our work. We utilized L2 distance (which is equivalent to cosine similarity for normalized embeddings) because prior work has established it as the standard, robust metric for measuring semantic similarity in high-dimensional deep representation spaces.
> Prior work shows that performance can be further improved by explicitly selecting “diverse” neighborhoods [3]. Overall, we nevertheless find that even the simple choice of L2 distance leads to robust performance gains of TTT over global training across architectures and modalities.
>
> > Line 239-241: The Authors claim that the sparse TTT classifier is functionally equivalent based on 89% match (except for pathological cases). Could the authors elaborate on what "pathological" means? 11% seems to be relatively high and I don't understand where should I expect these two would deviate.
>
> Thank you for pointing out this ambiguity. By "pathological" we refer to test samples located in regions of the feature space characterized by poor neighborhood quality (e.g., high class variance or low density). In these regions, the support set retrieved for Test-Time Training (TTT) lacks a clear semantic signal, causing the decision rules of both sparse and dense models to become unstable. To support this, we analyzed the model confidence (probability of the Top-1 class) for both the matching (89%) and diverging (11%) cases:
>
> * High confidence in the matching case: In the 89% of cases where the models agree, both models exhibit high confidence (0.80−0.95). This indicates that for coherent neighborhoods (low class variance), both models effectively utilize the local structure to form a reliable decision boundary.
> * Low confidence in the divergent case: In the 11% of cases where the models disagree, the average confidence drops significantly to below 38%.
>
> To break down the divergent case (11%) even further, we observed one of the models (sparse SAE TTT or dense CLIP TTT) is correct in approximately 56% of the cases with confidence values in the range of 0.32–0.38. In the remaining 44% of the cases, both models fail to predict the correct class with even lower confidence scores (0.25−0.30).
>
> Conclusion: The 11% deviation does not represent a failure of the sparse model to capture key features. Rather, it represents noisy cases where the local data manifold is ambiguous. In these "pathological" scenarios, neither model finds a strong decision rule, leading to arbitrary disagreements.
>
> ---
>
> Thank you again for your detailed review. Please let us know in case you have any remaining questions or concerns.

---

> > ### Author Response · Authors · 2025-11-22
> >
> > References:
> >
> > [1] Delétang et al. Language Modeling Is Compression. arXiv:2309.10668. 2023
> >
> > [2] Carlini et al. Quantifying Memorization Across Neural Language Models. arXiv:2202.07646. 2022
> >
> > [3] Hübotter et al. Efficiently Learning at Test-Time: Active Fine-Tuning of LLMs. arXiv:2410.08020. 2024

---

> > > ### Comment · Reviewer_zPcj · 2025-11-24
> > >
> > > Thank you for answering my questions. These answers reassured me in my initial positive rating of this work. Thus, I will happily recommend acceptance of the work.

---

> > > > ### Author Response · Authors · 2025-11-26
> > > >
> > > > Thank you for your follow-up and for re-evaluating our work. We are glad to hear that the additional references and experiments have reassured you in your positive position on this work. We appreciate your support.

---

### Official Review · Reviewer_fGSc · 2025-11-02

**Soundness:** 3
**Presentation:** 3
**Contribution:** 3
**Rating:** 6
**Confidence:** 3

**Summary:**

The authors conduct a comprehensive investigation into the potential of test-time training (TTT) to reduce in-distribution test error by leveraging only previously seen data. While TTT has traditionally been employed to address distribution shifts, this work explores its applicability to in-distribution scenarios. The authors first demonstrate that TTT encourages the model to converge to sparse solutions in the concept space. Building on this insight, they empirically show that TTT can effectively reduce in-distribution error rates on MNIST, ImageNet, and language modeling.

**Strengths:**

- The idea of leveraging test-time training (TTT) to reduce in-distribution error is both novel and well-motivated, particularly given that most prior work on TTT has focused on addressing distribution shifts. The authors indeed demonstrate that TTT effectively reduces in-distribution test error compared to standard global training.
- The analysis of the model’s representation space, grounded in the linear representation hypothesis, provides a solid conceptual foundation for understanding how models learn and encode semantic concepts.
- The paper presents a rigorous theoretical analysis of the in-distribution test error behavior under TTT, which strengthens the overall contribution and supports the empirical findings.

**Weaknesses:**

- I believe the term “foundation model” requires clearer definition in the context of this paper. As shown in Figure 4, the effectiveness of test-time training (TTT) in reducing in-distribution error appears to diminish as model capacity increases. This observation raises concerns about the applicability of TTT to large-scale foundation models, which are typically defined as models with significantly larger parameter counts (e.g., exceeding 32 billion parameters). This concern is also acknowledged in Takeaway 1. However, in the introduction, the authors state that foundation models are underparameterized, which seems to contradict the commonly accepted understanding of the term. This inconsistency may lead to confusion for readers, and I believe a more precise and consistent definition of foundation model would help clarify the paper’s contributions and claims.
- Although not a major concern, the experiment using MNIST with LeNet-5 appears somewhat outdated. While it is understandable that the authors aimed to demonstrate their method on a simple dataset and model, the use of a more recent dataset and architecture would have strengthened the empirical evaluation, particularly in light of the paper’s claims regarding foundation models.

**Questions:**

Please refer to my weaknesses.

---

> ### Author Response · Authors · 2025-11-22
>
> Thank you for your review, and the positive evaluation of our work. We are happy to address each point.
>
> > The idea of leveraging test-time training (TTT) to reduce in-distribution error is both novel and well-motivated, particularly given that most prior work on TTT has focused on addressing distribution shifts. The authors indeed demonstrate that TTT effectively reduces in-distribution test error compared to standard global training.
>
> Thank you!
>
> > I believe the term “foundation model” requires clearer definition in the context of this paper. As shown in Figure 4, the effectiveness of test-time training (TTT) in reducing in-distribution error appears to diminish as model capacity increases. This observation raises concerns about the applicability of TTT to large-scale foundation models, which are typically defined as models with significantly larger parameter counts (e.g., exceeding 32 billion parameters). This concern is also acknowledged in Takeaway 1. However, in the introduction, the authors state that foundation models are underparameterized, which seems to contradict the commonly accepted understanding of the term. This inconsistency may lead to confusion for readers, and I believe a more precise and consistent definition of foundation model would help clarify the paper’s contributions and claims.
>
> Thank you for raising this important point! We agree that the term “foundation model” commonly refers to models of massive scale (multiple billion parameters) that are trained on vast and diverse datasets. When we describe foundation models as underparameterized, we are referring to their capacity relative to the vast complexity of their training distributions (e.g., the entire internet). Despite having billions of parameters, these models are effectively lossy compressors of knowledge [1]. They cannot perfectly memorize the entire training set [2], which is where Test-Time Training (TTT) provides value by adapting the model to the specific instance at test-time.
>
> The notion that foundation models are “underparameterized” is tightly connected to extensive results on scaling laws [e.g., 3, 4]. Intuitively speaking, a model might be seen as “underparameterized” if further scaling of model size leads to improved performance. Prior theoretical work has found, for example, that smooth interpolation of training data requires $n \cdot d$ parameters where $d$ is the ambient data dimension [5], indicating that even “seemingly large” models might be insufficiently large to model large and complex datasets.
>
> We clarified the connection of underparameterization and foundation models with an additional sentence in the introduction. Thank you again for raising this important point.
>
> In this work, **we observe the known phenomenon of scaling/underparameterization** and **we provide evidence on how, when, and why test-time training can improve performance under this phenomenon**. We believe that our results extensively validate our hypotheses across scales (from MNIST & ImageNet to multi-billion parameter language models) and are likely to extend to even larger foundation models on internet-scale datasets. However, training such foundation models lies beyond our compute constraints.
>
> > Although not a major concern, the experiment using MNIST with LeNet-5 appears somewhat outdated. While it is understandable that the authors aimed to demonstrate their method on a simple dataset and model, the use of a more recent dataset and architecture would have strengthened the empirical evaluation, particularly in light of the paper’s claims regarding foundation models.
>
> Although MNIST is an older benchmark, we included it deliberately to illustrate the universality of the phenomenon across scale and architectural paradigms. Our experiments span simple convolutional networks on MNIST to modern Transformer architectures trained on ImageNet and The Pile (Section 4). Anchoring our evaluation with MNIST alongside massive datasets confirms that the phenomenon is invariant to model size and data complexity. This confirms that our results are not artifacts of specific architectures or foundation models, but rather reflect a fundamental characteristic that persists from the simplest baselines to the largest scales.
>
> ---
>
> Thank you again for your detailed review. Please let us know in case you have any remaining questions or concerns.
>
> [1] Delétang et al. Language Modeling Is Compression. arXiv:2309.10668. 2023
>
> [2] Carlini et al. Quantifying Memorization Across Neural Language Models. arXiv:2202.07646. 2022
>
> [3] Kaplan, et al. Scaling laws for neural language models. arXiv preprint arXiv:2001.08361, 2020.
>
> [4] Hoffmann, et al., Training Compute-Optimal Large Language Models. arXiv preprint arXiv:2203.15556, 2022.
>
> [5] Bubeck and Sellke. A universal law of robustness via isoperimetry. In NeurIPS, 2021.

---

> > ### Comment · Reviewer_fGSc · 2025-11-26
> >
> > I have reviewed the author’s response and find it reasonable. Among the points raised, I believe it would be beneficial to provide a clearer definition of the term “foundation model” within the paper. Based on this, I will maintain my current score. Thank you for your detailed responses.

---

> > > ### Author Response · Authors · 2025-11-26
> > >
> > > Thank you for your feedback! We agree that a clearer definition is beneficial and have updated the first paragraph of the Introduction to explicitly define foundation models as "large-scale neural networks trained on vast datasets that can be applied across a wide range of use cases" (Bommasani et al., 2021).
> > >
> > > Do these changes fully address your concerns?

---

### Author Response · Authors · 2025-12-02
**Summary of Discussion Period**

Dear Area Chair,

Below we would like to succinctly summarize the changes of our revision during the discussion period (highlighted in red in the PDF).

Main changes:
* We conducted an additional experiment on MNIST using end-to-end TTT rather than training only the linear head. The results are analogous to when only training the linear head (v2NY, V2Zv) supporting our hypothesis that TTT learns the local meaning of concepts.
* We expanded the MNIST scaling study to include larger models. This further supports our hypothesis that TTT improvements diminish as models become overparameterized (V2Zv).
* Slight improvements to presentation (following suggestions from reviewers fGSc, 1Gcj, v2NY).
* We listed additional literature to support notions of sparse concept spaces and the linear representation hypothesis (v2NY).

In our view, with our revisions, we have adeptly addressed all remaining concerns of reviewers.

Overall, we are excited that reviews are in unanimous support of acceptance, highlighting our work's novelty (fGSc, V2Zv), rigor (fGSc, zPcj, 1Gcj, v2NY), significant relevance (zPcj, v2NY, V2Zv), and that the paper is well-written (zPcj, v2NY, V2Zv).

---

### Meta-Review · Area_Chair_AdB8 · 2026-01-07

**Summary:**

All five reviewers agree on acceptance (fGSc: 6, v2NY: 6, zPcj: 8, V2Zv: 6, 1Gcj: 8) with varying confidence. Although confidence was lacking for the highest scores, the reviews still made specific points, and informed the decision alongside the reading of the submission itself by the meta-reviewer. The main arguments for acceptance are the (1) focus on in-distribution and already seen data that complements the focus on out-of-distribution data in existing work on test-time training, (2) the variety and scope of experiments from toy to standard and on vision and text, (3) empirical and theoretical content as much of the work on test-time training/adaptation/updates has been more empirical. The main weaknesses raised were about exposition, which were addressed by clarification and revision, and the testing of assumptions or exploration of results when assumptions do not hold, which were addressed by reference and experiment to support the chosen assumptions. The meta-reviewer agrees with the reviewer consensus on acceptance.

**Reviewer Concerns:**

- restricting adaptation to linear updates / last layer parameters (V2Zv): addressed by end-to-end experiment in the toy MNIST setting, though not addressed by doing the precise experiment asked for in the larger setting (updating the last two layers of larger models), but ultimately this is addressed on the balance
- imprecise language and insufficient background (fGSc, 1Gcj): addressed by revising the introduction and in the response which provided further explanation and the highlighting of references

**Reviewer Scores:**

- fGSc: maintain 6 or raise to 7 given clarification of terminology ("foundation model") and discussion of the toy setting of MNIST for some but not all of the results
- zPcj: maintain 8 given overall positivity and detailed response by authors
- 1Gcj: maintain 8 given positivity and self-identified unfamiliarity with the topic (the clarification by the authors should only support the positive non-expert evaluation through its clarification)
- v2NY: maintain 6 and not raise given knowledge of the topic and complete response by the authors—though complete there is simply a reasonable difference of opinion of the assumptions and how much experiments should violate the assumptions
- V2Zv: maintain 6 or raise to 7 as the author response provides new experiments as requested that help to confirm the empirical and theoretical results which are reflected in the appendices of the submission

---

### Decision · Program_Chairs · 2026-01-26

Accept (Poster)